# A Skewness-Based Criterion for Addressing Heteroscedastic Noise in Causal Discovery

**Yingyu Lin**[*1], **Yuxing Huang**[*2], **Wenqin Liu**[*3], **Haoran Deng**[*4], **Ignavier Ng**[5],
**Kun Zhang**[5,6], **Mingming Gong**[3,6], **Yi-An Ma**[1], **Biwei Huang**[1]
[1]UC San Diego, [2]New York University, [3]The University of Melbourne, [4]Zhejiang University,
[5]Carnegie Mellon University, [6]Mohamed bin Zayed University of Artificial Intelligence
`{yil208,yianma,bih007}@ucsd.edu,yh6004@nyu.edu,`
`{wenqin.liu,mingming.gong}@unimelb.edu.au,denghaoran@zju.edu.cn,`
`{ignavierng,kunz1}@cmu.edu`

## Abstract

Real-world data often violates the equal-variance assumption (homoscedasticity), making it essential to account for heteroscedastic noise in causal discovery. In this work, we explore heteroscedastic symmetric noise models (HSNMs), where the effect $Y$ is modeled as $Y = f(X) + \sigma(X)N$, with $X$ as the cause and $N$ as independent noise following a symmetric distribution. We introduce a novel criterion for identifying HSNMs based on the skewness of the score (i.e., the gradient of the log density) of the data distribution. This criterion establishes a computationally tractable measurement that is zero in the causal direction but nonzero in the anticausal direction, enabling the causal direction discovery. We extend this skewness-based criterion to the multivariate setting and propose `SkewScore`, an algorithm that handles heteroscedastic noise without requiring the extraction of exogenous noise. We also conduct a case study on the robustness of `SkewScore` in a bivariate model with a latent confounder, providing theoretical insights into its performance. Empirical studies further validate the effectiveness of the proposed method.

## 1 Introduction

Discovering causal relationships among variables from data, known as causal discovery, is of great interest in many fields such as biology (Sachs et al., 2005) and Earth system science (Runge et al., 2019). The primary approaches to causal discovery include constraint-based methods (Spirtes & Glymour, 1991; Spirtes et al., 2001), which rely on conditional independence tests, and score-based methods (Heckerman et al., 1995; Chickering, 2002; Huang et al., 2018), which optimize a certain objective function. These methods often identify the causal structure only up to Markov equivalence (Spirtes et al., 2001; Glymour et al., 2019), since the causal directions in general cannot be determined without prior knowledge or additional assumptions (Pearl, 2009).

Another line of approaches, based on functional causal models, impose restrictions on the causal mechanisms to make the causal directions identifiable. The key idea is to show that, under these restrictions, no forward and backward models that result in independent noise can co-exist, which leads to an asymmetry that helps identify the causal direction. Examples include linear non-Gaussian models (Shimizu et al., 2006), nonlinear additive noise models (Hoyer et al., 2008a), and post-nonlinear causal models (Zhang & Hyvärinen, 2009). Most of these models assume that the noise terms are homoscedastic, i.e., they have constant variances across different samples. This assumption may limit their applicability because the noise terms are often heteroscedastic in real-world data, such as those in environmental science (Merz et al., 2021) and robotics (Kersting et al., 2007).

To address this limitation, there is growing interest in developing more general functional causal models capable of inferring causal directions despite the presence of heteroscedastic noise. Previous approaches for handling heteroscedastic noise typically involve extracting noise terms through nonlinear regressions to estimate conditional means and variances, followed by independence tests on the residuals (Strobl & Lasko, 2023; Immer et al., 2023). This procedure needs to be repeated for

---

*Equal contribution.

each variable, which can be computationally intensive. Alternative strategies that avoid extracting noise terms include the likelihood-based approach, which often requires the noise to follow Gaussian distributions (Immer et al., 2023; Khemakhem et al., 2021; Duong & Nguyen, 2023), and the kernel-based criterion proposed by Mitrovic et al. (2018), whose theoretical guarantees for handling heteroscedastic noise may not be clear.

In this work, we introduce a novel criterion to address symmetric heteroscedastic noise in causal discovery. This is motivated by the fact that symmetric noise encompasses a wide range of widely used assumptions in noise distributions, such as Gaussian, centered uniform, Laplace, and Student's t distributions, with applications in, e.g., finance (Anderson & Arnold, 1993), environmental science (Damsleth & El-Shaarawi, 2018), and psychology (McGill, 1962). Specifically, we leverage the skewness of the score (the gradient of the log density) of the data distribution to establish a measurement that is zero in the causal direction but positive in the anticausal direction, thereby uncovering the causal direction. We extend this skewness-based criterion to the multivariate setting and propose `SkewScore`, an algorithm that effectively manages heteroscedastic noise without needing to extract the exogenous noise. This approach adopts the two-phase ordering-based search framework for DAGs introduced by Teyssier & Koller (2005). First, we estimate the topological order by iteratively identifying the sink node of the causal graph using the skewness-based criterion. Then, we prune the directed acyclic graph (DAG) associated with this topological order using classical methods, such as conditional independence tests (Zhang et al., 2011). This reduces the computational complexity, lowering the number of conditional independence tests from exponential to polynomial in the number of vertices.

Furthermore, we conduct a case study to evaluate the robustness of `SkewScore` in a bivariate model with a latent confounder. Existing approaches based on functional causal models typically do not allow latent confounders, as their presence will violate the independent noise condition required for causal direction identification. However, our theoretical insights show that in this bivariate model with a latent confounder, when the causal effect between the observed variables dominates that of the latent confounder, `SkewScore` can still correctly identify the causal direction. Additionally, our method enables the quantification of the latent confounder's impact within this triangular model.

## 2 RELATED WORK

**Heteroscedastic Noise Models (HNMs).** Causal discovery with HNMs has gained significant interest in recent years. HNMs relax the assumption of homoscedastic noise, which was generally assumed in previous studies. Xu et al. (2022) adopted a modeling choice where the variance of the scaled noise variable is a piecewise constant function of its parents, which may restrict the possible mechanisms of the variance. Tagasovska et al. (2020); Strobl & Lasko (2023); Immer et al. (2023) considered a more general class of HNMs, where the conditional variance is a deterministic function of its parents. The works by Tagasovska et al. (2020) and Immer et al. (2023) focus primarily on identifying pairwise causal relationships. Both Strobl & Lasko (2023) and Duong & Nguyen (2023) include a noise-extraction step to estimate the sink node, with Strobl & Lasko (2023) using mutual information and Duong & Nguyen (2023) using normality as criteria. Yin et al. (2024) propose a two-phase continuous optimization framework to learn the causal graph.

**Latent Confounder.** Causal discovery aims to identify causal relationships from observational data. However, traditional methods typically assume the absence of latent confounders in the causal graph, an assumption that often does not hold in real-world scenarios. To address this challenge, extensive research has focused on learning causal structures that allow latent variables. These approaches include methods based on conditional independence tests (Spirtes et al., 2001; Colombo et al., 2012; Akbari et al., 2021), over-complete ICA-based techniques (Hoyer et al., 2008b; Salehkaleybar et al., 2020), Tetrad condition (Silva et al., 2006; Kummerfeld & Ramsey, 2016), high-order moments (Shimizu et al., 2009; Cai et al., 2019; Xie et al., 2020; Adams et al., 2021; Chen et al., 2022), matrix decomposition techniques (Anandkumar et al., 2013), mixture oracles (Kivva et al., 2021) and rank constraints (Huang et al., 2022; Dong et al., 2023). Nevertheless, these methods either recover the causal graph only up to a broad equivalence class or assume that each latent confounder has at least two pure children that are not adjacent, which forbids the existence of triangle structure involving latent confounders, that is $X \leftarrow Z \rightarrow Y$ and $X \rightarrow Y$, where $Z$ is the latent variable.

**Causal Discovery with Score Matching.** A recent class of causal discovery algorithms employs constraints on the score, i.e., the gradient of the log density function, to uniquely identify the

causal graph in additive noise models (ANMs). This line of thought determines the topological order by iteratively identifying sink nodes (i.e., leaf nodes) using the score (Rolland et al., 2022; Montagna et al., 2023c; Sanchez et al., 2023; Montagna et al., 2023b). Interestingly, Montagna et al. (2023a) empirically indicates that the presence of latent confounders may not significantly disrupt the inference of the topological ordering; score-matching-based approaches still provide reliable ordering. Effective gradient estimation techniques are crucial in connecting the gradient of the log-likelihood to successful causal discovery. Various methods have been proposed to estimate the score from samples generated by an unknown distribution, including score matching (Hyvärinen & Dayan, 2005; Vincent, 2011; Song & Ermon, 2019; Song et al., 2020), and kernel score estimators based on Stein's methods (Li & Turner, 2017; Shi et al., 2018; Zhou et al., 2020). These methods have been successfully applied to generative modeling, representation learning, and addressing intractability in approximate inference algorithms.

# 3 PROBLEM SETUP AND PRELIMINARIES

**Symbols and notations.** Let $\mathbb{R}_+ = (0, \infty)$. We denote by $C^1(\mathbb{R}^k)$ the set of all continuously differentiable functions on $\mathbb{R}^d$, and by $C^1(\mathbb{R}^k, \mathbb{R}_+)$ the subset of $C^1(\mathbb{R}^k)$ consisting of functions that take only positive values. We denote by $L^\infty(\mathbb{R}^k)$ the set of all essentially bounded functions on $\mathbb{R}^k$. Furthermore, let $L^\infty := \cup_{k=1}^\infty L^\infty(\mathbb{R}^k)$, and $C^1 = \cup_{k=1}^\infty C^1(\mathbb{R}^k)$. For brevity, we write $\frac{\partial}{\partial x_i}$ as $\partial_{x_i}$. For a random variable $X = (X_1, X_2, \ldots, X_d)$ in $\mathbb{R}^d$, we represent $(X_1, \ldots, X_{i-1}, X_{i+1}, \ldots, X_d)$ as $X_{-i}$, and similarly, $(x_1, \ldots, x_{i-1}, x_{i+1}, \ldots, x_d)$ as $x_{-i}$. We use $p_X(x)$, or simply $p(x)$, to denote the probability density of $X$, and $p_{X_i|X_{-i}}(x_i|x_{-i})$ as $p(x_i|x_{-i})$ for conditional densities. We denote by $\mathrm{pa}(X_i)$ the set of parents (direct causes) of $X_i$ in a directed acyclic graph (DAG), and by $\mathrm{child}(X_i)$ the set of children (direct effects) of $X_i$. A node with no children, i.e., no outgoing edges, is called a *sink* or *leaf* node.

## 3.1 HETEROSCEDASTIC SYMMETRIC NOISE MODELS (HSNM)

We consider a structural causal model over $d$ random variables $\{X_i\}_{i=1}^d$, where each variable $X_i$ corresponds to a node in the directed acyclic graph (DAG) $G$. The Heteroscedastic Symmetric Noise Model (HSNM) is defined as

$$X_i = f_i(\mathrm{pa}(X_i)) + \sigma_i(\mathrm{pa}(X_i))N_i, \quad i = 1, \ldots, d, \tag{1}$$

where $N_i$ is an exogenous noise variable that follows a symmetric distribution with mean zero, i.e., $p_{N_i}(-n_i) = p_{N_i}(n_i)$. The noise terms $\{N_i\}_{i=1}^d$ are jointly independent, and each $N_i$ is independent of $\mathrm{pa}(X_i)$ when $\mathrm{pa}(X_i) \neq \emptyset$. We assume that $p_X \in C^1(\mathbb{R}^d, \mathbb{R}_+)$, and for each $i \in \{1, 2, \ldots, d\}$, $p_{N_i} \in C^1(\mathbb{R}, \mathbb{R}_+)$, with $f_i, \sigma_i \in C^1$ and $\sigma_i \in L^\infty$. Additionally, all partial derivatives $\partial_{x_j} \sigma_i \in L^\infty$. We also assume there exists a constant $r > 0$ such that $\inf_{i \in \{1, \ldots, d\}} \inf_x \sigma_i(x) \geq r$. Furthermore, we assume that $p_X, p_N, f_i$, and $\sigma_i$ satisfy weak regularity conditions to ensure that the derivatives and expectations involved in our analysis are well-defined.

It is worth noting that symmetric noise encompasses a wide range of widely used assumptions in noise distributions, such as Gaussian, centered uniform, Laplace, and Student's t distributions, with applications in, e.g., finance (Anderson & Arnold, 1993), environmental science (Damsleth & El-Shaarawi, 2018), and psychology (McGill, 1962).

## 3.2 SCORE FUNCTION, SCORE MATCHING, AND SKEWNESS

In this paper, the *score* function refers to the gradient of the log density, $\nabla p(x)$, which can be estimated using various score matching methods Hyvärinen & Dayan (2005); Vincent (2011); Song & Ermon (2019); Song et al. (2020); Li & Turner (2017); Shi et al. (2018); Zhou et al. (2020). We introduce two important properties of the score function and define the skewness of the score.

**Facts** *Let $p(x_1, x_2, \cdots, x_d) \in C^1(\mathbb{R}^d, \mathbb{R}_+)$ be a density function that is strictly postive on $\mathbb{R}^d$. The following two facts hold for every $i = 1, 2 \cdots, d$:*

**Fact 1.** $\partial_{x_i} \log p(x_i|x_1, \cdots, x_{i-1}, x_{i+1}, \cdots, x_d) = \partial_{x_i} \log p(x_1, x_2, \cdots, x_d)$.

**Fact 2.** $\mathbb{E}_{X \sim p}[\partial_{x_i} \log p(X)] = 0$.

The proofs of Fact 1 and Fact 2 are provided in Appendix D. These two facts are crucial for deriving Theorem 1, a skewness-based criterion for HSNMs. Fact 1 allows us to work with derivatives of the

log joint density rather than directly estimating conditional densities, while Fact 2 shows that the expectation of the score is a zero vector, when $p \in C^1(\mathbb{R}^d, \mathbb{R}_+)$.

Skewness measures the asymmetry of a probability distribution. In particular, Pearson's moment coefficient of skewness is defined as the normalized third central moment. For a real-valued random variable $W$, the unnormalized skewness is given by $Skew(W) = \mathbb{E}\left[(W - \mathbb{E}[W])^3\right]$. Inspired by this, we introduce the notion of skewness for the score function.

**Definition 1.** *Given a probability density $p$ for a random variable $X = (X_1, X_2 \cdots, X_d) \in \mathbb{R}^d$ with $p(x) \in C^1(\mathbb{R}^d, \mathbb{R}_+)$, we define its skewness of the score in the $x_i$-coordinate as:*

$$\begin{aligned} SkewScore_{x_i}(p) : &= \left| \mathbb{E}_{X \sim p} \left[ \left( \partial_{x_i} \log p(X) - \mathbb{E}_{X \sim p} \left[ \partial_{x_i} \log p(X) \right] \right)^3 \right] \right| \\ &= \left| \mathbb{E}_{X \sim p} \left[ \left( \partial_{x_i} \log p(X) \right)^3 \right] \right|, \end{aligned} \tag{2}$$

*where the second equality follows from Fact 2.*

This measure captures the asymmetry within each coordinate of the score. If the conditional density $p(x_i \mid x_{-i})$ is a symmetric function, then $SkewScore_{x_i}(p) = 0$, as discussed in Appendix A.

## 4 CAUSAL DISCOVERY WITH THE SKEWNESS-BASED CRITERION

In this section, we focus on identifying causal relationships within heteroscedastic symmetric noise models (HSNM), as formalized in Eq. (1). We introduce a novel identification criterion, the *skewness-of-score criterion*, which captures the asymmetry between the causal and anti-causal directions to determine the causal order in the infinite sample limit. We begin by providing intuitive explanations of this criterion, present theoretical guarantees for the bivariate causal model, and extend the results to multivariate cases. We then present an algorithm based on this criterion in Section 4.2.

### 4.1 IDENTIFICATION WITH THE SKEWNESS-BASED CRITERION

We begin with the bivariate case to build intuition for the skewness-based criterion. The extension to multivariate models follows naturally, as shown in Corollary 2.

Consider the following bivariate heteroscedastic symmetric noise model (HSNM):

$$Y = f(X) + \sigma(X)N, \tag{3}$$

where $N$ is symmetric exogenous noise independent of $X$ with mean zero, i.e., $p_n(-n) = p_n(n)$, and $X \perp\!\!\!\perp N$. We assume the same regularity conditions stated in Section 3.1, specified as Assumption 2 and 3 in Appendix E.

The key insight for the identifiability of the heteroscedastic symmetric noise model (HSNM), as given in Eq. (3), is that the conditional distribution $P(Y \mid X)$ is symmetric, while $P(X \mid Y)$ is generally asymmetric under mild assumptions on $f$ and $\sigma$. This contrast is illustrated in Figure 1. To quantify the asymmetry of these conditional distributions, a typical measure is the skewness, defined as the standardized central third moment. However, directly estimating the skewness of the *conditional distributions* from data is challenging due to the complexity of the conditional expectation.

To circumvent this challenge, we instead turn to an auxiliary random variable — the *score function* $\nabla \log p(X, Y)$, where $(X, Y)$ follows the data distribution $p$. Specifically, Fact 1 allows us to convert the partial derivatives of the log conditional density into the gradient of the log joint density, which is the score function. This transformation reframes the problem of analyzing the conditional distribution as one involving the score function, thereby simplifying the estimation process. Consequently, this leads to the identifiability criterion for heteroscedastic symmetric noise models, formally stated in Theorem 1, with the model restriction given in Assumption 1.

**Assumption 1** (Model Restriction). Denote $A(x) = \frac{p'_x(x)}{p_x(x)} - \frac{\sigma'(x)}{\sigma(x)}$, $B(x) = -\frac{f'(x)}{\sigma(x)}$, and $C(x) = \frac{\sigma'(x)}{\sigma(x)^2}$, where $p'_x$, $\sigma'$, and $f'$ represent the derivatives of $p_x$, $\sigma$, and $f$ respectively. Assume the following inequality holds:

$$\int \left[ \left( A(x) + C(x) \frac{p'_n(u)u}{p_n(u)} \right)^3 + 3 \left( B(x) \frac{p'_n(u)}{p_n(u)} \right)^2 \left( A(x) + C(x) \frac{p'_n(u)u}{p_n(u)} \right) \right] p_n(u) p_x(x) \mathrm{d}x \mathrm{d}u \neq 0.$$

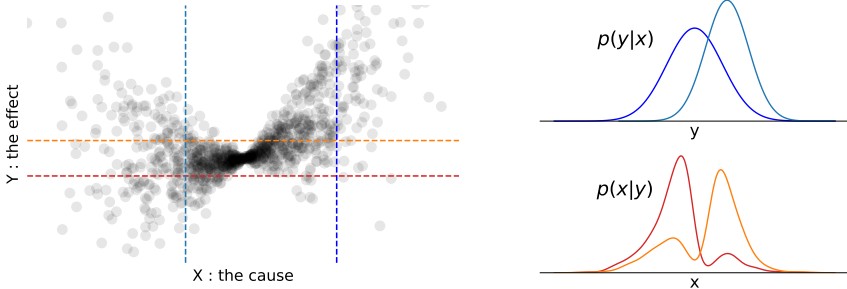

Figure 1: Identifiability of the heteroscedastic symmetric noise model (HSNM). For the causal direction $X \to Y$ (left), the conditional distribution $p(y|x)$ is symmetric for any $x$ (top right), and the variance of $p(y|x)$ varies with different values of $x$ due to heteroscedasticity. In contrast, $p(x|y)$ is asymmetric for some $y$ (bottom right).

Assumption 1 excludes the scenario where the skewnesses of the score's projections are both zero in the causal and anti-causal directions. In particular, it rules out linear Gaussian additive models. In Proposition 5 in Appendix B, we demonstrate that Assumption 1 holds generically by showing that the set of models violating this assumption has a property analogous to having a zero Lebesgue measure. More discussion is provided in Appendix B.

We now present a key result that shows how skewness of the score of the data distribution provides useful information that helps uncover the causal direction.

**Theorem 1.** *Let $(X, Y)$ follows the heteroscedastic symmetric noise model given by Eq. (3). Let $p(x, y)$ denote the joint distribution of $(X, Y)$. The following asymmetry holds.*

1. *The score function's component in the $Y$-coordinate (the effect direction) follows an unskewed distribution:*

$$\text{SkewScore}_y(p) := \left| \mathbb{E}_{X,Y} \left[ (\partial_y \log p(X, Y))^3 \right] \right| = 0. \tag{4}$$

2. *Moreover, under Assumption 1, the score function's component in the $X$-coordinate (the cause direction) follows a skewed distribution:*

$$\text{SkewScore}_x(p) := \left| \mathbb{E}_{X,Y} \left[ (\partial_x \log p(X, Y))^3 \right] \right| \neq 0. \tag{5}$$

The proof is provided in Appendix E. The asymmetry in the above theorem can be used to determine the causal direction between $X$ and $Y$. Specifically, one could estimate the score function $\nabla \log p(x, y)$ and determine the causal direction based on the magnitude of the skewness of its $x$- and $y$-coordinate. This result naturally generalizes to the multivariate case for $d \geq 2$, as stated in the following corollary: the score's component only along the sink-node coordinate is unskewed, under weak assumptions.

We first clarify the notations: Let $X = (X_1, X_2, \cdots X_d) \in \mathbb{R}^d$ follow the multivariate heteroscedastic symmetric noise model given by Eq. (1), and $p(x) = p(x_1, x_2, \cdots, x_d)$ denote the joint distribution of $X$. For simplicity, denote $\mathrm{d}x = \mathrm{d}x_1 \mathrm{d}x_2 \cdots \mathrm{d}x_d$, $f_i = f_i(\text{pa}_i(x))$, $\sigma_i = \sigma_i(\text{pa}_i(x))$, $p_i = p_{N_i}\left(\frac{x_i - f_i}{\sigma_i}\right)$ and $p_i' = p_{N_i}'\left(\frac{x_i - f_i}{\sigma_i}\right)$.

**Corollary 2.** *Assume that the following condition holds for every node $k$ with direct causes (i.e., $\text{child}(k) \neq \emptyset$):*

$$\int_{\mathbb{R}^d} \left[ \frac{p_k'}{\sigma_k p_k} - \sum_{i \in \text{child}(k)} \left( \frac{p_i'}{p_i} \cdot \frac{\sigma_i \partial_{x_k} f_i + (x_i - f_i)\partial_{x_k}\sigma_i}{\sigma_i^2} + \frac{\partial_{x_k}\sigma_i}{\sigma_i} \right) \right]^3 \prod_{j=1}^d \frac{p_j}{\sigma_j} \mathrm{d}x \neq 0. \tag{6}$$

*Then, for every node $j$, it is a sink if and only if the score's component in the $x_j$-coordinate follows an unskewed distribution, i.e.,*

$$\text{SkewScore}_{x_j}(p) = 0,$$

*where $\text{SkewScore}_{x_j}(p)$ is defined in Definition 1.*

The proof is provided in Appendix F. This corollary enables us to find the sink nodes of a DAG with the knowledge of the score function. By iteratively applying this process, we can determine the topological order of the graph. In Section 4.2, we provide a practical algorithm that leverages this Corollary for identifying the causal order.

## 4.2 ALGORITHM

We apply Corollary 2 and propose an algorithm to determine the causal ordering, named Skewscore (Algorithm 1). Our approach follows the topological order search framework of Rolland et al. (2022), which determines the topological order by iteratively identifying sink nodes. Different from their method, we identify the node with the minimum skewness of score as the sink node in each iteration. Compared to existing methods, Skewscore circumvents the need to extract the exogenous noise (Immer et al., 2023; Duong & Nguyen, 2023; Montagna et al., 2023b) or to estimate the diagonal of the Hessian of the log density (Rolland et al., 2022; Sanchez et al., 2023). Additionally, after finding the minimum $SkewScore_{x_j}$ in Line 6 of Algorithm 1, we can check whether $SkewScore_{x_j}$ is close to zero, where a value not close to zero indicates a violation of the symmetric noise assumption or other model assumptions. We note that the algorithm's numerical stability and sample complexity can depend on the test odd function chosen in the skewness definition, see Appendix C.

Once the topological order is obtained using Algorithm 1, the complete DAG can be constructed by adding edges with conditional independence tests (Zhang et al., 2011) associated with the topological order. In this order-based search, $\mathcal{O}(d^2)$ CI tests are performed, compared to exponential complexity in classical methods (Spirtes & Glymour, 1991) in worst-case scenarios.

---

**Algorithm 1** SkewScore

1: **Input:** Data matrix $\mathbf{X} \in \mathbb{R}^{n \times d}$, test odd function $\psi(s) = s^3$, conditional independence test oracle CI (returns p-value), and significance level $\alpha \in (0, 1)$.
2: **Initialization:** nodes $= \{1, \ldots, d\}$, topological order $\pi = []$, adjacency matrix $A \in \mathbb{R}^{d \times d}$ (all zeros).
3: **for** $k = 1, \ldots, d$ **do**
4:      Estimate the score function $\mathbf{s}(\mathbf{X}) = \nabla \log P_{\mathbf{X}_{\text{nodes}}}(\mathbf{X})$.
5:      Estimate $SkewScore_{x_j} = |\mathbb{E}_{\mathbf{X}_{\text{nodes}}}[\psi(\mathbf{s}_j(\mathbf{X}))]|$.    ▷ Skewness of the score's projection on $x_j$
6:      $\ell \leftarrow \arg\min_{j \in \text{nodes}} SkewScore_{x_j}$             ▷ Find the sink node
7:      $\pi \leftarrow [\ell, \pi]$                         ▷ Update the topological order
8:      nodes $\leftarrow$ nodes $- \{\ell\}$
9:      Remove $\ell$-th column of $\mathbf{X}$
10: **end for**
11: **for** $i = 1, \ldots, d - 1$ **do**
12:      **for** $j = i + 1, \ldots, d$ **do**
13:          **if** $\text{CI}(X_{\pi_i}, X_{\pi_j}, X_{\pi_{\{1, \ldots, j-1\} \setminus \{i\}}}) < \alpha$ **then**
14:             $A_{\pi_i, \pi_j} = 1$                  ▷ Add edge $\pi_i \rightarrow \pi_j$
15:          **end if**
16:      **end for**
17: **end for**
18: **Output:** Topological order $\pi$, and adjacency matrix $A$.

---

# 5 CASE STUDY: ROBUSTNESS TO THE LATENT CONFOUNDER IN A BIVARIATE SETTING

It has been shown that various types of assumption violations may affect the performance of causal discovery (Reisach et al., 2021; Montagna et al., 2023a; Ng et al., 2024). In this section, we conduct a case study to analyze the robustness of Algorithm 1 to the latent confounder in a pairwise causal model given by Eq. (7). We provide theoretical insights into this robustness in Proposition 3, with experimental results presented in Section 6.

We focus on a three-variable model consisting of two observed variables, $X$ and $Y$, and one latent variable, $Z$, as depicted in Figure 2. The model is formally expressed as:

$$X = \phi_0(Z) + N_0,$$
$$Y = \lambda \tilde{f}(X) + \phi_1(Z) + \sigma(X)N_1, \tag{7}$$

where $N_i \sim q_i$ with $q_i(n) = q_i(-n)$ and $q_i \in C^1(\mathbb{R}, \mathbb{R}_+)$, for $i = 0, 1$, and $Z \sim p_z(z)$. In this model, the effect of the latent confounder has an additive structure, and the conditional standard deviation $\sigma(X)$ only depends on the observed variable $X$, without interaction with the latent variable. The function $f$ is replaced with $\lambda \tilde{f}$, where $\lambda > 0$ scales the effect of $X$ on $Y$. To analyze the confounding effects of $Z$, we vary $\lambda$ while keeping $\tilde{f}, \sigma, \phi_i, q_i$, and $p_z(z)$ fixed. As in Section 3.1, we assume $\tilde{f}, \sigma \in C^1(\mathbb{R})$, $\sigma, \sigma' \in L^\infty(\mathbb{R})$, and that there exists a constant $r > 0$ such that $\sigma(x) \geq r$.

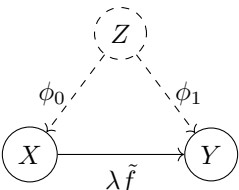

Figure 2: Three-variable HSNM model with latent confounder given by Eq. (7).

The criterion in Theorem 1 identifies the causal order without requiring exogenous-noise extraction. This simplicity allows us to quantify the impact of latent confounders. Notably, to recover the causal direction in the infinite sample limit, Line 6 in Algorithm 1 is equivalent to observing that the score's component in the effect coordinate is less skewed than in the cause. The following proposition shows that in Model (7) with $\lambda$ large enough, the causal direction $X \to Y$ can be identified by Algorithm 1 in the infinite sample limit, with only mild regularity assumptions on the function $\tilde{f}, \phi_i$, $q_i$ for $i \in \{0, 1\}$, and $p_z$. These conditions are discussed in Appendix H along with the proof.

**Proposition 3.** *Suppose two variables $X$ and $Y$ follow the causal models in Eq.(7). Given $\tilde{f}, \sigma, \phi_i$, $q_i$ for $i \in \{0, 1\}$, and $p_z$, let $p_\lambda(x, y)$ denote the joint distribution of $(X, Y)$. Under assumption 4, there exists $\lambda^* \in \mathbb{R}_+$, such that for every $\lambda \geq \lambda^*$, we have*

$$SkewScore_x(p_\lambda) > SkewScore_y(p_\lambda),$$

*where $SkewScore_x(\cdot), SkewScore_y(\cdot)$ is defined in Definition 1. This indicates the effect $Y$ can be identified in Line 6 in Algorithm 1 in the infinite sample limit.*

Roughly speaking, when $\lambda$ is sufficiently large, similar to a high signal-to-noise ratio (SNR), the effect of the latent variable $Z$ can mostly be treated as noise. In this setting, $\lambda$ serves as an indicator of how much the observed variables dominate over the latent confounder. The threshold $\lambda^*$ is the critical value beyond which the causal influence of $X$ on $Y$ becomes strong enough that the skewness-based measure of the causal direction $X \to Y$ consistently surpasses that of the reverse direction $Y \to X$. For $\lambda < \lambda^*$, this skewness-based measure may not clearly distinguish the causal direction. However, once $\lambda$ exceeds $\lambda^*$, the log-density derivatives of $X$ show much higher skewness, allowing for the reliable identification of $Y$ as the effect variable, even when there is a latent confounder.

For the multivariate extension, a notable limitation is that the DAG recovery method in Lines 11-17 of Algorithm 1 may lead to spurious direct links when there are latent confounders. A potential way to mitigate this issue is to leverage the idea from FCI (Spirtes et al., 2001), which will involve more sophisticated ways of performing conditional independence tests (after obtaining the topological ordering) and output the Partial Ancestral Graph (PAG) instead of just a DAG.

Finally, we present an explicit analysis of a simplified homoscedastic case of the triangular model defined by Eq. (7), where the exogenous noises and the cause follow Gaussian distributions, and the effects of the latent variable $Z$ on both $X$ and $Y$ are assumed to be linear.

**Example 4.** *Consider the following model:*

$$\begin{cases} X & = Z + N_0, \\ Y & = Z + f(X) + N_1, \end{cases}$$

*where $Z, N_i$ for $i \in \{0, 1\}$ are independent variables, all distributed as $\mathcal{N}(0, 1)$. Then, we have:*

$$\begin{cases} Skew_x & = \frac{2\sqrt{2\pi}}{\pi} \left| \int_{\mathbb{R}} x f'(x)(1 + f'(x)) e^{-\frac{x^2}{2}} \, dx \right|, \\ Skew_y & = 0. \end{cases} \tag{8}$$

*The explicit calculation of Eq. (8) can be found in Appendix I. If f is a linear function, then $Skew_x = 0$ and the model is unidentifiable by Algorithm 1. However, in most cases, $Skew_x > 0$, making the model identifiable by Algorithm 1. For example, for a quadartic function $f(x) = ax^2 + bx + c$ with $a \neq 0$, $Skew_x > 0$ if and only if $b \neq \frac{1}{2}$. Similarly, for almost all polynomials with a degree greater than 2, we have $Skew_x > 0$, indicating that Algorithm 1 can effectively determine the causal direction in the infinite sample limit.*

## 6 EXPERIMENTS

**Setup.** We compare `SkewScore` to state-of-the-art causal discovery methods to demonstrate the effectiveness of our proposed methods. We conduct experiments using synthetic data generated from HSNMs in three settings: (1) bivariate HSNMs; (2) latent-confounded triangular HSNM setting (bivariate model with a latent confounder, as discussed in Section 5); and (3) multivariate HSNMs. For each scenario, noise is sampled from two types of symmetric distributions: Gaussian distribution and Student's *t* distribution. The degree of freedom of the Student's *t* distribution is uniformly sampled from $\{2, 3, 4, 5\}$. For the multivariate setting, the causal conditional mean function *f* is modeled by a Gaussian process with a radial basis function (RBF) kernel (unit bandwidth), while the conditional standard deviation, $\sigma$, is parameterized using a sigmoid function. In both the bivariate and triangular settings, two data generation procedures are used: one with the Gaussian process-sigmoid setup (denoted as "GP-sig"), the same procedure as the multivariate setting, and another with *f* modeled as an invertible sigmoid function and $\sigma$ set to the absolute function (denoted as "Sig-abs"). The causal graphs are constructed via the Erdös-Rényi (ER) model, where for a given number of nodes *d*, the average number of edges is also set to *d*, for $d \geq 2$. The sample size for all experiments is 5000. The score estimator employs sliced score matching (SSM) with a 3-layer MLP. We use a batch size of 128 and configure the hidden dimensions to 128 for $d < 10$ and 512 for $d \geq 10$. Optimization is performed using the Adam optimizer with a learning rate of $10^{-3}$, and we subsample 1000 points for conditional independence testing after obtaining the topological order.

**Comparisons.** We empirically evaluate `SkewScore` against state-of-the-art bivariate causal discovery methods designed for heteroscedastic noise models: HECI (Xu et al., 2022); LOCI (Immer et al., 2023) and HOST (Duong & Nguyen, 2023). Given that Since HOST can also handle multivariate settings, we include it in those comparisons. We also test DiffAN (Sanchez et al., 2022), which uses the second-order derivative of the log-likelihood, estimated through diffusion models. Additionally, we evaluate NoTEARS (Zheng et al., 2018), which uses continuous optimization for linear DAG learning, and the MLP version of DAGMA (Bello et al., 2022), which handles nonlinear DAG learning. For methods that output a DAG, we compute the corresponding topological order. For baseline methods designed only for cause-effect pairs, we perform comparisons using data generated from pairwise causal relations with $d = 2$.

**Metrics.** For the pairwise causal discovery settings, we report the accuracy of the estimated causal directions. For multivariate cases, we compute two quantities: the topological order divergence (Rolland et al., 2022), and the structural Hamming distance (SHD) between the estimated and the ground truth graphs. The topological order divergence evaluates how well the estimated topological order aligns with the ground truth. Given an ordering $\pi$ and a binary ground-truth adjacency matrix $A$, the topological order divergence is defined as $D_{\text{top}}(\pi, A) = \sum_{i=1}^{d} \sum_{j:\pi_i > \pi_j} A_{ij}$. Lower values of $D_{\text{top}}(\pi, A)$ indicate higher accuracy in the estimated topological order, reflecting more precise causal discovery. The structure Hamming distance measures the difference between two DAGs by counting the number of edge additions, deletions, or reversals required to transform one into the other.

**Results.** The results of the synthetic experiments for the bivariate, latent-confounded-triangular, and multivariate scenarios are shown in Figure 3a, Figure 3b and Figure 4, respectively. For the pairwise causal discovery task, the results are averaged over 100 independent runs, while for the multivariate task, they are averaged over 10 runs. Our approach, `SkewScore`, consistently outperforms or matches other methods across the bivariate, latent-confounded triangular, and multivariate settings. In pairwise causal discovery tasks (Figure 3), methods designed for heteroscedastic noise models (HNMs) generally perform well, but tend to make more errors when confounders are present or when the noise follows a Student's *t* distribution. In contrast, methods like NoTEARS, DAGMA, and DiffAN—which are not specifically designed for HNMs—face performance limitations in both pairwise and multivariate scenarios. `SkewScore` demonstrates consistently strong performance in the latent-confounded triangular setting, supported by the theoretical insights discussed in Section 5.

All HNM-based methods perform well in the latent-confounded setting with the Sig-abs structural causal model formulation, where LOCI handles both Gaussian and Student's $t$-distributed noise effectively, and HOST performs well for the Gaussian noise case. However, when the latent-confounded data is generated using the GP-sig formulation, these HNM baselines do not match the performance of `SkewScore`. In more challenging multivariate scenarios, our method outperforms the others, especially as the dimensionality increases. HOST depends on noise extraction and normality tests, which can be difficult with complex data. DiffAN performs well with a small number of variables, but its performance drops as dimensionality grows. This is likely due to its non-retraining approach and the score derived from ANM, which may lead to error accumulation over iterations. Additionally, all experiments are run on CPUs, though our method can easily be deployed on GPUs. This suggests the potential for further scalability by leveraging modern computational advancements. More experiments and discussions are provided in Appendix J.

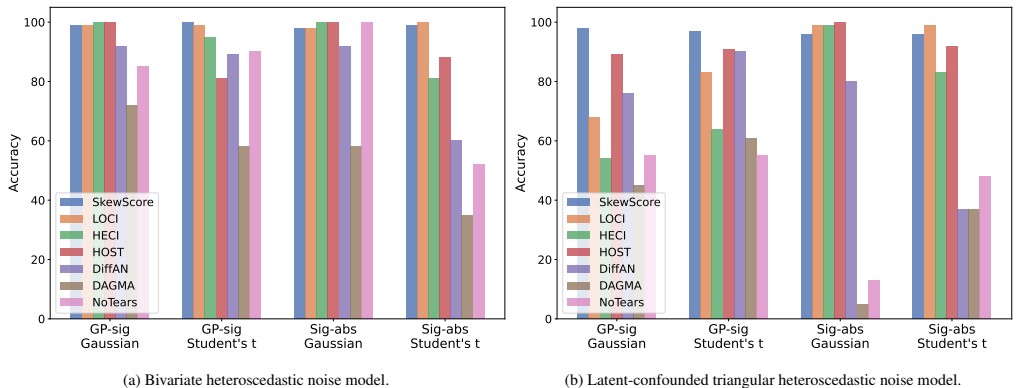

(a) Bivariate heteroscedastic noise model.

(b) Latent-confounded triangular heteroscedastic noise model.

Figure 3: Accuracy of causal direction estimation across different data generation processes for (a) bivariate heteroscedastic noise models, and (b) latent-confounded triangular heteroscedastic noise models. Results are averaged over 100 independent runs.

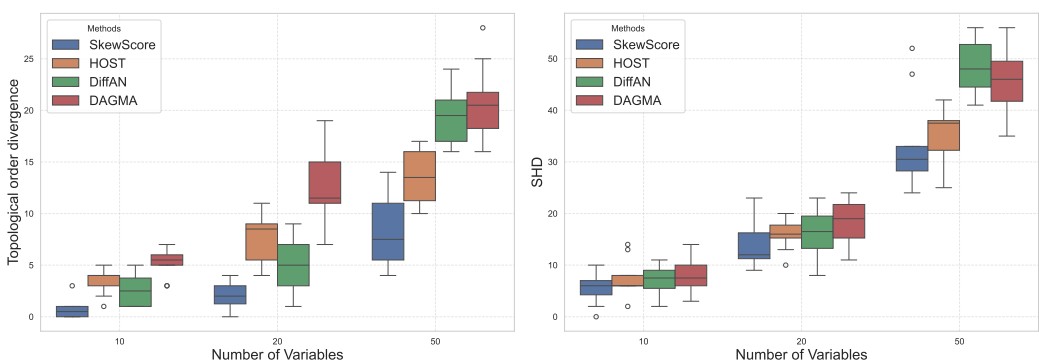

Figure 4: Topological order divergence and structural Hamming distance (SHD) across different number of variables (the dimension $d$). Lower values indicate better performance for both metrics.

## 7 CONCLUSION

In this study, we introduced a novel criterion—the skewness-of-score criterion—for causal discovery with heteroscedastic symmetric noise models. Leveraging this criterion, we developed the algorithm `SkewScore`, which effectively identifies causal directions without the need for extracting exogenous noise. We also conduct a case study on the robustness of our algorithm in a triangle latent-confounded model, providing theoretical insights into its performance under this setup. Theoretical guarantees and empirical validations support our methodology. Future work will focus on extending the method to extremely high-dimensional cases, where the number of variables and the complexity of their interactions significantly increase. Additionally, future research will explore causal discovery in causal models involving multiple latent confounders and more general latent structures.

## ACKNOWLEDGEMENTS

The research is partially supported by the NSF awards: SCALE MoDL-2134209, CCF-2112665 (TILOS). It is also supported by the U.S. Department of Energy, the Office of Science, the Facebook Research Award, as well as CDC-RFA-FT-23-0069 from the CDC's Center for Forecasting and Outbreak Analytics. BH is supported by NSF DMS-2428058. We would also like to acknowledge the support from NSF Award No. 2229881, AI Institute for Societal Decision Making (AI-SDM), the National Institutes of Health (NIH) under Contract R01HL159805, and grants from Quris AI, Florin Court Capital, and MBZUAI-WIS Joint Program. YL thanks Sumanth Varambally for the discussion. YH thanks Xuyang Lin for the discussion.

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

## A  ADDITIONAL DISCUSSION ON THE *SkewScore* MEAUSURE

This section discusses *SkewScore* (Definition 1), including intuition and motivation for considering this measure.

If the conditional density $p(x_i \mid x_{-i})$ is a symmetric function, then we have $SkewScore_{x_i}(p) = 0$. The proof of this result is deferred to the end of this section. For asymmetric distributions, we provide two one-dimensional examples with explicit computations at the end of this section:

1. For the Gumbel distribution $p_{\text{Gumbel}}(x) = \frac{1}{\beta} \exp\left(-\frac{x-\mu}{\beta} - e^{-\frac{x-\mu}{\beta}}\right)$ with $\beta > 0$, we have $SkewScore(p_{\text{Gumbel}}) = \frac{2}{\beta^3}$.

2. For the Gamma distribution $\Gamma(k, \theta)$, with $k > 3$, and $\theta > 0$, i.e.,

$$p_{\text{Gamma}}(x) = \begin{cases} \frac{1}{\Gamma(k)\theta^k} x^{k-1} e^{-\frac{x}{\theta}}, & x \geq 0 \\ 0, & x < 0 \end{cases} \quad (\theta > 0, k > 3),$$

we have $SkewScore(p_{\text{Gamma}}) = \theta^{-3} \frac{4}{(k-3)(k-2)}$.

We provide additional discussion on the motivation for considering the skewness of the score, i.e., the *SkewScore* measure. Consider the bivariate HSNM model where the variable $X$ causes $Y$, i.e., $X \to Y$. Figure 1 shows that to identify the causal direction, rather than examining the skewness of the residuals, we can focus on the skewness of the conditional distribution. However, working directly with the conditional random variables $X|Y$ and $Y|X$ is often challenging. Therefore, we introduce an auxiliary random variable, $\nabla \log p(X, Y)$, where $(X, Y)$ follows the data distribution $p$. This two-dimensional random variable, induced by the score function on the data distribution, is represented as the black directed line segments in Figure 5. This auxiliary random vector's first element, $\partial_x \log p(X, Y)$, is visualized as the orange directed line segment, while the second element, $\partial_y \log p(X, Y)$, is shown in blue. Fact 1 connects the conditional distribution $p(y|x)$ and $p(x|y)$ with this auxiliary random vector, allowing us to transform the conditional-distribution-estimation problem to a joint-distribution one. Theorem 1 shows that under Assumption 1, the orange directed segment (representing $\partial_x \log p(X, Y)$) follows a skewed distribution, whereas the blue directed segment (representing $\partial_y \log p(X, Y)$) follows a symmetric distribution. This means that $SkewScore_x(p) \neq 0$ while $SkewScore_y(p) = 0$. We summarize the motivation in Figure 6.

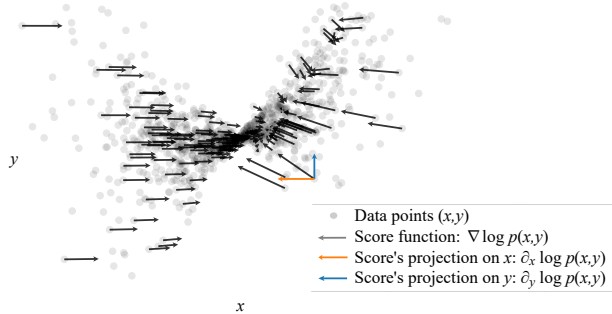

Figure 5: Score function (a vector field) visualized on data points with projections on the $x$-axis and $y$-axis. For clarity, only the projections for one data point are shown (highlighted in orange and blue). Theorem 1 examines the skewness of these projections across the data distribution.

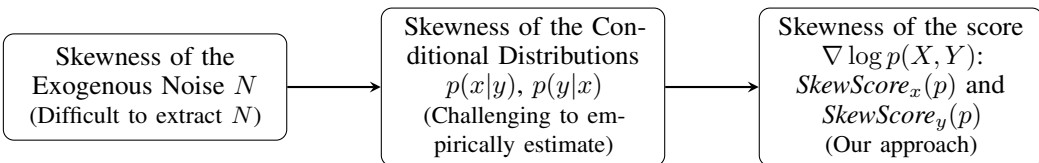

Figure 6: Conceptual flow from skewness of exogenous noise to skewness of the score.

We present the deferred proof and derivation as follows.

**Proof of:** *SkewScore*$_{x_i}(p) = 0$ **for symmetric** $p(x_i \mid x_{-i})$**.** By Definition 1, we have

$$SkewScore_{x_i}(p) = \left| \mathbb{E}_{X \sim p} \left[ (\partial_{x_i} \log p(X))^3 \right] \right|$$

$$= \left| \int_{\mathbb{R}^{d-1}} \int_{\mathbb{R}} \frac{[\partial_{x_i} p(x_1, \cdots, x_d)]^3}{p(x_1, \cdots, x_d)^2} \mathrm{d}x_i \mathrm{d}x_{-i} \right|$$

The assumption that $p(x_i \mid x_{-i})$ is symmetric implies: for every $x \in \mathbb{R}^d$,

$$p(x_1, \cdots, x_i, \cdots, x_d) = p(x_1, \cdots, -x_i, \cdots, x_d).$$

Therefore,

$$\partial_{x_i} p(x_1, \cdots, x_i, \cdots, x_d) = -\partial_{x_i} p(x_1, \cdots, -x_i, \cdots, x_d).$$

Now we fix $x_{-i} \in \mathbb{R}^{d-1}$, and let $h(x_i) = \frac{[\partial_{x_i} p(x_1, \cdots, x_i, \cdots, x_d)]^3}{p(x_1, \cdots, x_i, \cdots, x_d)^2}$. From the previous analysis, we derive that $h(x_i) = -h(-x_i)$. So $h$ is an odd function and thus

$$\int_{\mathbb{R}} h(x_i) \mathrm{d}x_i = 0.$$

Consequently, for every $x_{-i} \in \mathbb{R}^{d-1}$,

$$\int_{\mathbb{R}} \frac{[\partial_{x_i} p(x_1, \cdots, x_d)]^3}{p(x_1, \cdots, x_d)^2} \mathrm{d}x_i = 0.$$

$$\mathbb{E}\left[ \frac{\partial}{\partial x_i} \log p(X_1, X_2, \cdots, X_d) \right] = \int_{\mathbb{R}^d} \frac{\frac{\partial}{\partial x_i} p(x_1, x_2, \cdots, x_d)}{p(x_1, x_2, \cdots, x_d)} p(x_1, x_2, \cdots, x_d) \mathrm{d}x_{-i} \mathrm{d}x_i$$

$$= \int_{\mathbb{R}^{d-1}} \int_{\mathbb{R}} \frac{\partial}{\partial x_i} p(x_1, x_2, \cdots, x_d) \mathrm{d}x_i \mathrm{d}x_{-i}.$$

Thus we prove *SkewScore*$_{x_i}(p) = 0$. ■

**Derivation of *SkewScore*$(p_{\text{Gumbel}})$.** Recall that $p(x) = \frac{1}{\beta} \exp\left(-\frac{x-\mu}{\beta} - e^{-\frac{x-\mu}{\beta}}\right)$. Then $\log p(x) = \log\left(\frac{1}{\beta}\right) - \left(\frac{x-\mu}{\beta} + e^{-\frac{x-\mu}{\beta}}\right)$, and $\frac{d}{dx} \log p(x) = -\frac{1}{\beta} - \left(-\frac{1}{\beta}\right) e^{-\frac{x-\mu}{\beta}} = \frac{1}{\beta}\left(-1 + e^{-\frac{x-\mu}{\beta}}\right)$.

$$SkewScore(p_{\text{Gumbel}}) = \int_{\mathbb{R}} \left[ \frac{1}{\beta}\left(-1 + e^{-\frac{x-\mu}{\beta}}\right) \right]^3 \frac{1}{\beta} \exp\left(-\frac{x-\mu}{\beta} - e^{-\frac{x-\mu}{\beta}}\right) dx$$

$$\overset{z=\frac{x-\mu}{\beta}}{=} \frac{1}{\beta^3} \int_{\mathbb{R}} \left(-1 + e^{-z}\right)^3 e^{-(z+e^{-z})} dz$$

$$\overset{y=e^{-z}}{=} \frac{1}{\beta^3} \int_0^{\infty} \left(-1 + y\right)^3 e^{-y} dy.$$

Finally, since $\int_0^{\infty} y^k e^{-y} dy = \Gamma(k+1) = k!$, for $k \in \mathbb{N}$, we conclude that $SkewScore(p) = \frac{2}{\beta^3}$. ■

**Derivation of *SkewScore*$(p_{\text{Gamma}})$.** We first note that $k > 3$ implies that $p(x) \in C^1$ and ensures the integrals involved are all well-defined. We have $\log p(x) = (k-1)\ln x - \frac{x}{\theta} + \ln\left(\frac{1}{\Gamma(k)\theta^k}\right)$, and thus $\frac{d}{dx} \log p(x) = \frac{k-1}{x} - \frac{1}{\theta}$. Then we have

$$SkewScore(p) = \mathbb{E}\left[ \left( \frac{d}{dx} \log p(x) \right)^3 \right]$$

$$= \int_0^{\infty} \left( \frac{k-1}{x} - \frac{1}{\theta} \right)^3 \frac{1}{\Gamma(k)\theta^k} x^{k-1} e^{-\frac{x}{\theta}} dx$$

$$= \frac{1}{\Gamma(k)\theta^k} \int_0^{\infty} \left[ (k-1)^3 x^{-3} - 3(k-1)^2 \frac{1}{\theta} x^{-2} + 3(k-1) \frac{1}{\theta^2} x^{-1} - \frac{1}{\theta^3} \right] x^{k-1} e^{-\frac{x}{\theta}} dx$$

$$= \frac{\theta^{k-3}}{\Gamma(k)\theta^k} \left[ (k-1)^3 \Gamma(k-3) - 3(k-1)^2 \Gamma(k-2) + 3(k-1)\Gamma(k-1) - \Gamma(k) \right].$$

By the property of Gamma function $\Gamma(z+1) = z\Gamma(z)$, we have

$$\mathbb{E}\left[\left(\frac{d}{dx}\log p(x)\right)^3\right] = \frac{1}{\Gamma(k)\theta^k}\Gamma(k)\theta^{k-3}\left[\frac{(k-1)^3}{(k-1)(k-2)(k-3)} - \frac{3(k-1)^2}{(k-1)(k-2)} + \frac{3(k-1)}{k-1} - 1\right]$$

$$= \theta^{-3}\frac{4}{(k-3)(k-2)}.$$

■

## B    DISCUSSION ON MODEL ASSUMPTION 1

The following proposition demonstrates that Assumption 1 is not overly restrictive and can generally be satisfied; specifically, the set of models that violate this assumption is relatively small. To establish this, we fix $p_x$, $p_n$, and $\sigma(x)$, and consider $f$ within a finite-dimensional space with a sufficiently large dimension to capture the features of the data. We then demonstrate that the set of functions $f(x)$ that do not satisfy the assumption has zero measure.

**Proposition 5.** *Fix $p_x$, $p_n$, and $\sigma(x)$. Consider $f$ within an $M$-dimensional linear space $V$. In this space $V$, define the non-compliant set as*

$$S_{\mathrm{nc}}(V) := \{f \in V : f \text{ does not satisfy Assumption 1}\}.$$

*With the notation in Assumption 1, define*

$$\begin{cases} h(x) := 3\left(\int_{\mathbb{R}} \frac{p_n'(u)^2}{p_n(u)}\mathrm{d}u\right)\frac{A(x)p_x(x)}{\sigma(x)^2} + 3\left(\int_{\mathbb{R}} \frac{p_n'(u)^3 u}{p_n(u)^2}\mathrm{d}u\right)\frac{C(x)p_x(x)}{\sigma(x)^2}, \\ c := \int \left(A(x) + C(x)\frac{p_n'(u)u}{p_n(u)}\right)^3 p_n(u)p_x(x)\mathrm{d}x\mathrm{d}u, \end{cases}$$

*which do not involve $f$.*

*Since $\dim V = M$, $V$ is isomorphic to $\mathbb{R}^M$ and thus naturally inherits the Lebesgue measure on $\mathbb{R}^M$. This induced measure does not depend on the choice of basis and is denoted by $m$. Assume there exists $x_0 \in \mathbb{R}$ such that $h(x_0) \neq 0$, then the following holds.*

*1. If $c \neq 0$, then the non-compliant set has zero measure, i.e.,*

$$m(S_{\mathrm{nc}}(V)) = 0.$$

*2. If $c = 0$, assume $V$ is the space of polynomial functions of degree at most $M-1$, i.e., $V = \left\{\sum_{i=0}^{M-1} a_i x^i : a_0, a_2, \ldots, a_{M-1} \in \mathbb{R}\right\}$. Assume $\int_{\mathbb{R}} \left(p_x(x)^2 + p_x'(x)^2\right) e^{\mu x^2}\mathrm{d}x < \infty$, for some $\mu \in (0, +\infty)$. Then there exists $M_0 \in \mathbb{N}$, such that for every $M \geq M_0$, the non-compliant set has zero measure, i.e.,*

$$m(S_{\mathrm{nc}}(V)) = 0.$$

This proposition indicates that Assumption 1 is generally satisfied for most models within a sufficiently large finite-dimensional space, as the set of functions violating the assumption has zero measure. The proof is provided in Appendix G.

**Remark 6.** *We note that when $\sigma(x)$ is not constant (i.e., in the heteroscedastic model), $c \neq 0$ in most cases. However, if $\sigma(x)$ is constant and $p_x$ is symmetric, then $c = 0$. In this scenario, if $X$ is a Gaussian variable, the assumption in the second case is always satisfied; that is, there exists some $\mu \in (0, +\infty)$ such that $\int_{\mathbb{R}} \left(p_x(x)^2 + p_x'(x)^2\right) e^{\mu x^2}\mathrm{d}x < \infty$.*

For the multivariate case, intuitively, the general validity of the assumption in Corollary 2 is similar to Proposition 5, where the set of functions $(f_i, \sigma_i)$ that violate Eq. (6) exhibits a property analogous to having Lebesgue zero measure. Specifically, we parameterize the causal model by parameterizing functions $f_i^\theta$ and $\sigma_i^\theta$, where $\theta$ belongs to the parameter space $\Theta \subset \mathbb{R}^K$ (e.g., a neural network; see Example 7). The left-hand side of Eq. (6) defines a function of the parameter $\theta$, and we write it as $F(\theta)$. With this parametrization, the model that violates Eq. (6) corresponds to the solution of the equation $F(\theta) = 0$. Under regular assumptions on $f_i$, $\sigma_i$, $p_X$, and $p_{N_i}$, $F$ is a real-analytic function. Then by applying (Mityagin, 2015, Proposition 0), the violated set $S_{\mathrm{nc}}(\Theta) = \{\theta \in \Theta : F(\theta) = 0\}$

has zero measure as long as the function $F$ is not a zero constant function. It is worth noting, however, that in the homoscedastic Gaussian linear model, $F$ is identically zero, and thus the proposition from Mityagin (2015) no longer applies.

We provide the following examples to give a more intuitive understanding of Assumption 1.

**Example 7.** *In this example we consider a Homoscedastic model, where $\sigma(x) = \sigma$ is a constant. Let $f$ be a two-layer Neural Network with Softplus activation function:*

$$f(x) = w_2 \log(1 + e^{(w_1 x + b_1)}) + b_2, \ w_1, w_2, b_1, b_2 \in \mathbb{R}.$$

*Assuming $p_x$ is an unimodal distribution symmetric with respect to $0$, then Assumption 1 is violated if and only if $w_1 = 0$ or $w_2 = 0$, i.e., when $f$ is a constant function.*

**Proof** The LHS of Assumption 1 equals to

$$\frac{3w_1^2 w_2^2}{\sigma^2} \int_{\mathbb{R}} \frac{1}{(1 + e^{-(w_1 x + b)})^2} p_x'(x) dx \int_{\mathbb{R}} \frac{(p_N'(u))^2}{p_N(u)} du.$$

Notice that $\frac{(p_N'(u))^2}{p_N(u)}$ is always non-negative, so $\int_{\mathbb{R}} \frac{(p_N'(u))^2}{p_N(u)} du > 0$. We only need to prove $\int_{\mathbb{R}} \frac{1}{(1 + e^{-(w_1 x + b)})^2} p_x'(x) dx \neq 0$ when $w_1 \neq 0$.

Notice that $p_x(x)$ is an even function thus $p_x'(x)$ is an odd function, thus

$$
\begin{aligned}
\int_{\mathbb{R}} \frac{p_x'(x)}{(1 + e^{-(w_1 x + b)})^2} dx &= \int_0^\infty \frac{p_x'(x)}{(1 + e^{-(w_1 x + b)})^2} p_x'(x) dx + \int_{-\infty}^0 \frac{p_x'(x)}{(1 + e^{-(w_1 x + b)})^2} dx \\
&= \int_0^\infty \left( \frac{1}{(1 + e^{-(w_1 x + b)})^2} - \frac{1}{(1 + e^{-(-w_1 x + b)})^2} \right) p_x'(x) dx \\
&= \int_0^\infty \frac{(2 + e^{-(w_1 x + b)} + e^{-(-w_1 x + b)}) e^{-b} (e^{w_1 x} - e^{-w_1 x})}{(1 + e^{-(w_1 x + b)})^2 (1 + e^{-(-w_1 x + b)})^2} p_x'(x) dx
\end{aligned}
$$

Since $w_1 \neq 0$, and $p_x$ is an unimodal distribution symmetric with respect to $0$. The integrand is always of the same sign (always strictly greater than $0$ or strictly less than $0$) over the integration interval $(0, \infty)$. Therefore, the above integral is non-zero.

∎

**Example 8.** *In this example we consider the Heteroscedastic case with $\sigma(x) = e^{\lambda(x - \mu)^2}$, where $\mu \neq 0$, $\lambda > 0$. Let $f(x) = ax + b$, $X \sim \mathcal{N}(0, 1)$, $N \sim \mathcal{N}(0, 1)$. Then $f$ violates Assumption 1 if and only if*

$$a = \pm \sqrt{-\frac{c}{\gamma}},$$

*where $\gamma = 3\mu\lambda \left[ \frac{4 e^{\frac{\mu^2}{2(4\lambda+1)}}}{(4\lambda+1)\sqrt{4\lambda+1}} + \frac{6 e^{\frac{\mu^2}{2(6\lambda+1)}}}{(6\lambda+1)\sqrt{6\lambda+1}} \right]$ and $c$ is defined in Proposition 5. Especially, when $\frac{c}{\gamma} > 0$, $f$ always satisfy Assumption 1.*

It is evident that the non-compliant set $S_{\mathrm{nc}}(V) = \{\sqrt{-\frac{c}{\gamma}}\} \times \mathbb{R} \cup \{-\sqrt{-\frac{c}{\gamma}}\} \times \mathbb{R}$ always has $0$ measure (it is an empty set when $\frac{c}{\gamma} < 0$), providing an example for proposition 5. Additionally, due to the simple structure, we already have $m(S_{\mathrm{nc}}(V)) = 0$ when $c = 0$. Therefore, the discussion of the second case in Proposition 5 is unnecessary. The proof of this example is provided as follows.

**Proof** Recall in the proof of proposition 5 (Eq. (16)), we proved that $f$ violates Assumption 1 equals to

$$\int_{\mathbb{R}} f'(x)^2 h(x) \mathrm{d}x = -c.$$

As $f(x) = ax + b$ is linear, it equals to

$$a^2 \int_{\mathbb{R}} h(x) \mathrm{d}x = -c.$$

On the other hand, we can simplify $h(x)$ as follows:

$$h(x) = \frac{3p'_X(x)}{\sigma(x)^2} - \frac{9p_X(x)\sigma(x)}{\sigma(x)^4} = -\frac{3x}{\sqrt{2\pi}}e^{-\frac{4\lambda(x-\mu)^2+x^2}{2}} - \frac{18\lambda(x-\mu)}{\sqrt{2\pi}}e^{-\frac{6\lambda(x-\mu)^2+x^2}{2}}.$$

Thus we can compute that

$$\int_{\mathbb{R}} h(x)\mathrm{d}x = 3\mu\lambda\left[\frac{4e^{\frac{\mu^2}{2(4\lambda+1)}}}{(4\lambda+1)\sqrt{4\lambda+1}} + \frac{6e^{\frac{\mu^2}{2(6\lambda+1)}}}{(6\lambda+1)\sqrt{6\lambda+1}}\right].$$

Then we know that $\int_{\mathbb{R}} h(x)\mathrm{d}x \neq 0$. Therefore, we conclude that $f$ violates Assumption 1 if and only if

$$a^2 = -\frac{c}{\int_{\mathbb{R}} h(x)\mathrm{d}x},$$

which equals to

$$a = \pm\sqrt{-\frac{c}{\gamma}}.$$

∎

In the following example, we discuss a case where $f$ does not satisfy Assumption 1.

**Example 9.** *Let $\sigma(x) = \sigma$ be a constant and $f$ be linear, then $f$ does not satisfy Assumption 1 if and only if:*

$$\int_{\mathbb{R}} \frac{p'_x(x)^3}{p_x(x)^2}\mathrm{d}x = 0. \tag{9}$$

When $X$ follows a symmetric distribution, Eq. (9) holds. Specifically, when $X$ follows a Gaussian distribution, Eq. (9) holds, implying that $f$ fails to satisfy Assumption 1. This indicates that the Gaussian Linear model is unidentifiable by our algorithm, consistent with previous research such as (Hoyer et al., 2008a). However, as illustrated in example 8, if $\sigma(x)$ is replaced by a non-constant function, the model becomes identifiable for most cases. The proof of this example is provided as follows.

**Proof** We prove it by directly simplifying the formula in Assumption 1. Notice that when $\sigma(x) = \sigma$ is a constant, $f(x) = ax + b$ is linear, $C(x) = 0$, $B(x) = -\frac{a}{\sigma}$, $A(x) = \frac{p'_x(x)}{p_x(x)}$. So $f$ does not satisfy Assumption 1 equals to

$$\int_{\mathbb{R}} \frac{p'_x(x)^3}{p_x(x)^2}\mathrm{d}x + \frac{3a^2}{\sigma^2}\int_{\mathbb{R}} \frac{p'_n(u)^2}{p_n(u)}\int_{\mathbb{R}} p'_x(x)\mathrm{d}x = 0.$$

Notice that $\int_{\mathbb{R}} p'_x(x)\mathrm{d}x = 0$, it then equals to

$$\int_{\mathbb{R}} \frac{p'_x(x)^3}{p_x(x)^2}\mathrm{d}x = 0,$$

thus completing the proof. ∎

## C    DISCUSSION ON THE SKEWNESS DEFINITION

To enhance the flexibility of our algorithm, we redefine the skewness of a random variable $W$ as $Skew_\psi(W) = \mathbb{E}[\psi(W - \mathbb{E}W)]$, employing a nonlinear odd test function $\psi$ that satisfies $\psi(-s) = -\psi(s)$. Specifically, in Theorem 1, the chosen function is $\psi(s) = s^3$. Note that for any symmetric random variable $W$, $Skew_\psi(W) = 0$ for any odd function $\psi$, thus upholding the validity of Eq. (4) under this generalized definition. To ensure that a $\psi$-analog of Eq. (5) also holds in the anti-causal direction, it is essential to verify the $\psi$-analog of Assumption 1. A necessary condition for $\psi$ is nonlinearity due to Fact 2. The choice of the odd test function affects the robustness of the algorithm: an effective odd test function captures the asymmetry of the random variable while requiring less sample complexity.

## D   PROOFS OF SUPPORTING LEMMAS AND FACTS

**Proof of Fact 1**  Without loss of generality, it suffices to provide the proof when $p \in C^1(\mathbb{R}^2)$. Notice that the derivative of the log conditional density translates to the derivative of the log joint distribution, then

$$
\begin{aligned}
\frac{\partial}{\partial x_i} \log p(x_i | x_{-i}) &= \frac{\partial}{\partial x_i} \log \left( \frac{p(x_1, x_2, \cdots, x_d)}{p(x_{-i})} \right) \\
&= \frac{\partial}{\partial x_i} \left( \log p(x_1, x_2, \cdots, x_d) - \log p(x_{-i}) \right) \\
&= \frac{\partial}{\partial x_i} \log p(x_1, x_2, \cdots, x_d).
\end{aligned}
$$

∎

**Proof of Fact 2**

$$
\begin{aligned}
\mathbb{E} \left[ \frac{\partial}{\partial x_i} \log p(X_1, X_2, \cdots, X_d) \right] &= \int_{\mathbb{R}^d} \frac{\frac{\partial}{\partial x_i} p(x_1, x_2, \cdots, x_d)}{p(x_1, x_2, \cdots, x_d)} p(x_1, x_2, \cdots, x_d) \mathrm{d}x_{-i} \mathrm{d}x_i \\
&= \int_{\mathbb{R}^{d-1}} \int_{\mathbb{R}} \frac{\partial}{\partial x_i} p(x_1, x_2, \cdots, x_d) \mathrm{d}x_i \mathrm{d}x_{-i}.
\end{aligned}
$$

Since $p$ is continuously differentiable, we apply the Newton-Leibniz formula to $x_i$. For any $(x_1, \cdots, x_{i-1}, x_{i+1}, \cdots x_d) \in \mathbb{R}^{d-1}$, we have

$$
\int_{\mathbb{R}} \frac{\partial}{\partial x_i} p(x_1, \cdots, x_i, \cdots, x_d) dx_i = p(x_1, \cdots, x_i, \cdots, x_d)|_{x_i=-\infty}^{x_i=+\infty} = 0 - 0 = 0,
$$

where $\lim_{x_i \to +\infty} p(x_1, \cdots, x_i, \cdots, x_d) = \lim_{x_i \to -\infty} p(x_1, \cdots, x_i, \cdots, x_d) = 0$ because $p$ is a probability density. Therefore, we have shown that $\mathbb{E} \left[ \frac{\partial}{\partial x_i} \log p(X_1, X_2, \cdots, X_d) \right] = 0$.   ∎

We also note that the assumption $p_X(x) \in C^1(\mathbb{R}^d, \mathbb{R}_+)$ can be relaxed. For example, Fact 2 holds for uniform distribution on $[-1, 1]$ despite having singularities at $-1$ and $1$, as the densities at these two endpoints cancel out after the Newton-Leibniz step. More generally, Fact 2 still holds when $p$ is piecewise continuously differentiable with singularity points $w_1 < w_2 < \ldots < w_m$, and $\sum_{i=1}^{m} [p(w_i+) - p(w_i-)] = 0$.

Attentive readers may observe, however, that for the exponential distribution ($p_{\exp}(x) = \lambda e^{-\lambda x}$ for $x \geq 0$ and $0$ otherwise), Fact 2 does not hold. This is because of its singularity at $x = 0$, where $p_{\exp}(0-) = 0$ and $p_{\exp}(0+) = 1$.

## E   PROOF OF THEOREM 1

**Assumption 2** (Regularity assumption for $p_x, p_n$). $p_n, p_x \in C^1$,

1. $\int_{\mathbb{R}} \frac{|(p_n'(u))^3|}{p_n^2(u)} du < \infty$,

2. $\int_{\mathbb{R}} \left| \frac{p_x'(x)^3}{p_x(x)^2} \right| dx < \infty$, $\int_{\mathbb{R}} \frac{|(p_n'(u))^3 u^3|}{p_n^2(u)} du < \infty$.

**Assumption 3** (Regularity assumption for $f$).  $f \in C^1$,

$$
\int_{\mathbb{R}} \left| f'(x)^3 \right| p_x(x) \mathrm{d}x < \infty.
$$

**Remark 10.** *Assumption 2 and 3 ensures all the integrals we discuss here are well-defined.*

**Proof of Theorem 1**  We first prove Eq. (4). As $Y = f(X) + \sigma(X)N$, the density of $(X, Y)$ should be

$$
p(x, y) = p_x(x) p(y|x) = \frac{p_x(x)}{\sigma(x)} p_n \left( \frac{y - f(x)}{\sigma(x)} \right).
$$

$$\mathbb{E}_{X,Y} \left( \partial_y \log p(X,Y) \right)^3 = \int_{\mathbb{R}^2} \frac{p_x(x)}{\sigma(x)^4} \frac{\left( p'_n \left( \frac{y-f(x)}{\sigma(x)} \right) \right)^3}{p_n^2 \left( \frac{y-f(x)}{\sigma(x)} \right)} \mathrm{d}x \mathrm{d}y$$

$$= \int_{\mathbb{R}^2} \frac{p_x(x)}{\sigma(x)^3} \frac{(p'_n(u))^3}{p_n^2(u)} \mathrm{d}x \mathrm{d}u$$

$$= \int_{\mathbb{R}} \frac{p_x(x)}{\sigma(x)^3} \mathrm{d}x \int_{\mathbb{R}} \frac{(p'_n(u))^3}{p_n^2(u)} \mathrm{d}u$$

$$= 0.$$

For Eq. (5), we first expand it explicitly:

$$\mathbb{E}_{X,Y} \left( \partial_x \log p(X,Y) \right)^3 = \int_{\mathbb{R}^2} \left[ \frac{p'_x(x)}{p_x(x)} - \frac{\sigma'(x)}{\sigma(x)} - \frac{p'_n(u)}{p_n(u)} \left( \frac{f'(x)}{\sigma(x)} + \frac{\sigma'(x)u}{\sigma(x)^2} \right) \right]^3 p_x(x) p_n(u) \mathrm{d}x \mathrm{d}u \tag{10}$$

$$= \int_{\mathbb{R}} p_x(x) \int_{\mathbb{R}} \left[ A(x) + B(x) \frac{p'_n(u)}{p_n(u)} + C(x) \frac{p'_n(u)u}{p_n(u)} \right]^3 p_n(u) \mathrm{d}u \mathrm{d}x. \tag{11}$$

where $A(x) = \frac{p'_x(x)}{p_x(x)} - \frac{\sigma'(x)}{\sigma(x)}$, $B(x) = -\frac{f'(x)}{\sigma(x)}$ and $C(x) = \frac{\sigma'(x)}{\sigma(x)^2}$, as is denoted in Assumption 1. As $p_n(u) = p_n(-u)$, we derive $\int_{\mathbb{R}} \frac{p'_n(u)^3}{p_n(u)^2} du = \int_{\mathbb{R}} p'_n(u) du = 0$, so we can can expand Eq. (11) and get

$$\mathbb{E}_{X,Y} \left( \partial_x \log p(X,Y) \right)^3 = \int_{\mathbb{R}} p_x(x) \int_{\mathbb{R}} \left[ \left( A(x) + C(x) \frac{p'_n(u)u}{p_n(u)} \right)^3 \mathrm{d}x \tag{12} \right.$$

$$\left. + 3 \left( B(x) \frac{p'_n(u)}{p_n(u)} \right)^2 \left( A(x) + C(x) \frac{p'_n(u)u}{p_n(u)} \right) \right] p_n(u) \mathrm{d}u. \tag{13}$$

By assumption, $\mathbb{E}_{X,Y} \left( \partial_x \log p(X,Y) \right)^3 \neq 0$, thus completing the proof.

∎

## F    PROOF OF COROLLARY 2

**Proof of corollary 2**   As in prior work, we assume all integrals below are well-defined. We first compute that: for every $k \in \{1, 2, \cdots d\}$,

$$\partial_{x_k} \log p(x) = \frac{p'_k}{\sigma_k p_k} - \sum_{i \in child(k)} \left( \frac{p'_i}{p_i} \cdot \frac{\sigma_i \partial_{x_k} f_i + (x_i - f_i) \partial_{x_k} \sigma_i}{\sigma_i^2} + \frac{\partial_{x_k} \sigma_i}{\sigma_i} \right). \tag{14}$$

It suffices to notice that the joint distribution of $X$ can be written as:

$$p(x) = \prod_{i=1}^{d} p \left( x_i | \mathrm{pa}_i(x) \right)$$

$$\log p(x) = \sum_{i=1}^{d} \log p \left( x_i \mid \mathrm{pa}_i(x) \right)$$

$$= \sum_{i=1}^{d} \log \left( p_{N_i} \left( \frac{x_i - f_i \left( \mathrm{pa}_i(x) \right)}{\sigma_i(\mathrm{pa}_i(x))} \right) \right) - \sum_{i=1}^{d} \log \sigma_i(\mathrm{pa}_i(x)).$$

We then differentiate with respect to $x_k$ and directly derive Eq. (14).

We are now prepared to prove the first assertion. Considering a sink $X_k$, we know that $child(k) = \emptyset$. Therefore, by the formula in the first line of the proof:

$$\partial_{x_k} \log p(x) = \frac{p'_k}{\sigma_k p_k}.$$

Then we can compute:

$$\mathbb{E}_X \left(\partial_{x_k} \log p(X)\right)^3 = \int_{\mathbb{R}^d} \left[\frac{p'_{N_k}\left(\frac{x_k - f_k(\mathrm{pa}_k(x))}{\sigma_k(\mathrm{pa}_k(x))}\right)}{\sigma_k(\mathrm{pa}_k(x)) \cdot p_{N_k}\left(\frac{x_k - f_k(\mathrm{pa}_k(x))}{\sigma_k(\mathrm{pa}_k(x))}\right)}\right]^3 \prod_{i=1}^d p\left(x_i | \mathrm{pa}_i(x)\right) \mathrm{d}x$$

$$= \int_{\mathbb{R}^d} \frac{p(\mathrm{pa}_k(x))}{\sigma_k(\mathrm{pa}_k(x))^4} \cdot \frac{\left(p'_{N_k}\left(\frac{x_k - f_k(\mathrm{pa}_k(x))}{\sigma_k(\mathrm{pa}_k(x))}\right)\right)^3}{p_{N_k}^2\left(\frac{x_k - f_k(\mathrm{pa}_k(x))}{\sigma_k(\mathrm{pa}_k(x))}\right)} \prod_{i \neq k} p\left(x_i | \mathrm{pa}_i(x)\right) \mathrm{d}x$$

$$= \int_{\mathbb{R}^d} \frac{p(\mathrm{pa}_k(x))}{\sigma_k(\mathrm{pa}_k(x))^3} \cdot \frac{\left(p'_{N_k}(u)\right)^3}{p_{N_k}^2(u)} \mathrm{d}u \prod_{i \neq k} \left(p\left(x_i | \mathrm{pa}_i(x)\right) \mathrm{d}x_i\right)$$

$$= \int_{\mathbb{R}^{d-1}} \frac{p(\mathrm{pa}_k(x))}{\sigma_k(\mathrm{pa}_k(x))^3} \cdot \prod_{i \neq k} \left(p\left(x_i | \mathrm{pa}_i(x)\right) \mathrm{d}x_i\right) \int_{\mathbb{R}} \frac{\left(p'_{N_k}(u)\right)^3}{p_{N_k}^2(u)} \mathrm{d}u$$

$$= 0.$$

We obtain the last equality because $\int_{\mathbb{R}} \frac{\left(p'_{N_k}(u)\right)^3}{p_{N_k}^2(u)} du = 0$. More explicitly, due to the assumption that $p'_{N_k}(u)$ is symmetric, $\frac{\left(p'_{N_k}(u)\right)^3}{p_{N_k}^2(u)}$ is a odd function and thus its integral is 0.

We then prove the second assertion. Recall that

$$\partial_{x_k} \log p(x) = \frac{p'_k}{\sigma_k p_k} - \sum_{i \in child(k)} \left(\frac{p'_i}{p_i} \cdot \frac{\sigma_i \partial_{x_k} f_i + (x_i - f_i) \partial_{x_k} \sigma_i}{\sigma_i^2} + \frac{\partial_{x_k} \sigma_i}{\sigma_i}\right).$$

Therefore,

$$\mathbb{E}_X \left(\partial_{x_i} \log p(X)\right)^3 = \int_{\mathbb{R}^d} \left[\frac{p'_k}{\sigma_k p_k} - \sum_{i \in child(k)} \left(\frac{p'_i}{p_i} \cdot \frac{\sigma_i \partial_{x_k} f_i + (x_i - f_i) \partial_{x_k} \sigma_i}{\sigma_i^2} + \frac{\partial_{x_k} \sigma_i}{\sigma_i}\right)\right]^3 \prod_{j=1}^d \frac{p_j}{\sigma_j} \mathrm{d}x.$$

Thanks to our assumption (6), we derive that:

$$\mathbb{E}_X \left(\partial_{x_i} \log p(X)\right)^3 \neq 0,$$

thus completing the proof. ∎

## G PROOF OF PROPOSITION 5

**Lemma 11.** *If $\int_{\mathbb{R}} \left(p_x(x)^2 + p'_x(x)^2\right) e^{\mu x^2} \mathrm{d}x < \infty$ for some $\mu \in (0, +\infty)$, recalling*

$$h(x) := 3 \left(\int_{\mathbb{R}} \frac{p'_n(u)^2}{p_n(u)} \mathrm{d}u\right) \frac{A(x) p_x(x)}{\sigma(x)^2} + 3 \left(\int_{\mathbb{R}} \frac{p'_n(u)^3 u}{p_n(u)^2} \mathrm{d}u\right) \frac{C(x) p_x(x)}{\sigma(x)^2},$$

*then we have :*

$$\int_{\mathbb{R}} h(x)^2 e^{\mu x} \mathrm{d}x < \infty.$$

**Proof** As $A(x) = \frac{p'_x(x)}{p_x(x)} - \frac{\sigma'(x)}{\sigma(x)}$ and $C(x) = \frac{\sigma'(x)}{\sigma(x)^2}$ and $\sigma, \sigma'$ are bounded and $\sigma(x) \geq r > 0$, there exists a constant $b > 0$ s.t. $|A(x)| \leq \left|\frac{p'_x(x)}{p_x(x)}\right| + b$, $|C(x)| \leq b$. Notice that $\int_{\mathbb{R}} \frac{p'_n(u)^2}{p_n(u)} \mathrm{d}u$ and $\int_{\mathbb{R}} \frac{p'_n(u)^3 u}{p_n(u)^2} \mathrm{d}u$ are all constant, then there exists constant $c_1 > 0$ s.t.

$$|h(x)| \leq c_1 \left(p_x(x) + |p'_x(x)|\right).$$

Therefore

$$|h(x)|^2 \leq 2 c_1^2 \left(p_x(x)^2 + p'_x(x)^2\right),$$

$$\implies \int_{\mathbb{R}} h(x)^2 e^{\mu x} \mathrm{d}x \leq \int_{\mathbb{R}} \left(p_x(x)^2 + p'_x(x)^2\right) e^{\mu x^2} \mathrm{d}x < \infty.$$

∎

Defining

$$L^2(\mathbb{R}, e^{\mu x^2}) := \left\{ h \in L^2(\mathbb{R}) : \int_{\mathbb{R}} h(x)^2 e^{\mu x} \mathrm{d}x < \infty \right\},$$

lemma 11 equals to saying that $h \in L^2(\mathbb{R}, e^{\mu x^2})$. The following lemma is interesting as the integral is taken over $\mathbb{R}$ instead of a finite interval, and thus the Stone Weierstrass theorem cannot be applied.

**Lemma 12.** *For $h \in L^2(\mathbb{R}, e^{\mu x^2})$, if $\int_{\mathbb{R}} x^n h(x) dx = 0$ for every $n \in \mathbb{N}$, then $h = 0$ almost everywhere.*

**Proof** Without loss of generality, we assume $\mu = 1$. It is well known that $L^2(\mathbb{R}, e^{\mu x^2})$ is a Hilbert space. Then by knowledge of Hermite polynomials, there exist a series of polynomials $\{H_n\}_{n=1}^{\infty}$ which is a orthogonal basis of $L^2(\mathbb{R}, e^{\mu x^2})$. $\int_{\mathbb{R}} x^n h(x) dx = 0$ for every $n \in \mathbb{N}$ implies that $\int_{\mathbb{R}} H_n(x) h(x) dx = 0$ for every $n \in \mathbb{N}$, thus proving $h = 0$. ∎

**Proof of Proposition 5** For every finite dimensional subspace $V \subset V_0$, $f \in S_{\mathrm{nc}}(V)$ equals to

$$0 = \int \left[ \left( A(x) + C(x) \frac{p'_n(u)u}{p_n(u)} \right)^3 + 3 \left( B(x) \frac{p'_n(u)}{p_n(u)} \right)^2 \left( A(x) + C(x) \frac{p'_n(u)u}{p_n(u)} \right) \right] p_n(u) p_x(x) \mathrm{d}x \mathrm{d}u. \tag{15}$$

In the above equation, only $B(x)$ involves $f(x)$. So we can simplified Eq. (15) into

$$\int_{\mathbb{R}} f'(x)^2 h(x) \mathrm{d}x + c = 0, \tag{16}$$

where

$$\begin{cases} h(x) = 3 \left( \int_{\mathbb{R}} \frac{p'_n(u)^2}{p_n(u)} \mathrm{d}u \right) \frac{A(x) p_x(x)}{\sigma(x)^2} + 3 \left( \int_{\mathbb{R}} \frac{p'_n(u)^3 u}{p_n(u)^2} \mathrm{d}u \right) \frac{C(x) p_x(x)}{\sigma(x)^2}, \\ c = \int \left( A(x) + C(x) \frac{p'_n(u)u}{p_n(u)} \right)^3 p_n(u) p_x(x) \mathrm{d}x \mathrm{d}u. \end{cases}$$

Let $f_i$ ($1 \le i \le d$) be a basis of $V$, $f = \sum_i a_i f_i$, then Eq. (16) equals to

$$\sum_{1 \le i,j \le d} a_i a_j \int_{\mathbb{R}} f'_i(x) f'_j(x) h(x) \mathrm{d}x + c = 0.$$

Let $J_V(a_1, a_2 \cdots, a_n) := \sum_{1 \le i,j \le d} a_i a_j \int_{\mathbb{R}} f'_i(x) f'_j(x) h(x) \mathrm{d}x + c$. $J$ is a polynomial and thus a real analytic function. Define $KerJ_V := \{(a_1, a_2, \cdots, a_d) : J_V(a_1, a_2, \cdots, a_n) = 0\}$, then

$$f \in S_{\mathrm{nc}}(V) \iff (a_1, a_2, \cdots, a_n) \in KerJ_V.$$

We first consider the second case where $\int_{\mathbb{R}} \left( p_x(x)^2 + p'_x(x)^2 \right) e^{\mu x^2} \mathrm{d}x < \infty$ for some $\mu \in (0, +\infty)$. By lemma 11, $\int_{\mathbb{R}} h(x)^2 e^{\mu x} \mathrm{d}x < \infty$. If $\int_{\mathbb{R}} x^n h(x) dx = 0$ for every $n \in \mathbb{N}$, then by lemma 12, $h = 0$ almost everywhere. Notice that $h$ is also continuous, implying that $h(x) = 0$ for every $x \in \mathbb{R}$. This contradicts with our assumption on $h$. So we know that there exist $M_0$, s.t. $\int_{\mathbb{R}} x^{M_0} h(x) \mathrm{d}x \ne 0$. Therefore, for $M \ge M_0 + 1$, if we take $V$ to be the space of polynomials with degree smaller than $M$ and $f_i(x) = x^{i-1}$, then $J_V$ is a non-zero function. By basic knowledge of real analysis Mityagin (2015)[Proposition 0], $KerJ_V$ has zero measure. So $m(S_{\mathrm{nc}}(V)) = m(KerJ_V) = 0$

For the first case, if $c \ne 0$, then $J_V(0, 0, \cdots 0) = c \ne 0$. So $J_V$ is a non-zero function for every finite dimensional space $V \subset C^1(\mathbb{R})$. Similar to the analysis above, we derive $m(S_{\mathrm{nc}}(V)) = 0$ and thus completing the proof.

∎

# H  PROOF OF PROPOSITION 3

Let

$$h(x, v) := \int_{\mathbb{R}} p(z) q_0(x - \phi_0(z)) \sigma(x)^{-1} q_1 \left( \frac{v - \phi_1(z)}{\sigma(x)} \right) dz.$$

Under our assumption in 5, it is easy to verify that $h$ is well-defined and belong to $C^{1,1}(\mathbb{R}^2) \cap L^\infty$. Formally, we denote

$$
\begin{cases}
A_3 &= -\int_{\mathbb{R}^2} \frac{f'(x)^3 \partial_v h(x,v)^3}{h(x,v)^2} dxdv, \\
A_2 &= 3\int_{\mathbb{R}^2} \frac{f'(x)^2 \partial_v h(x,v)^2 \partial_x h(x,v)}{h(x,v)^2} dxdv, \\
A_1 &= -3\int_{\mathbb{R}^2} \frac{f'(x)\partial_v h(x,v)\partial_x h(x,v)^2}{h(x,v)^2} dxdv, \\
A_0 &= \int_{\mathbb{R}^2} \frac{\partial_x h(x,v)^3}{h(x,v)^2} dxdv.
\end{cases}
$$

The well-defineness of $A_i$ ($0 \le i \le 3$) will be verified in the following lemma.

**Assumption 4** (Restriction on Model with a Latent Confounder).

1. (Regularity assumption.) It exists an open subset $U \subset (0, +\infty)$ s.t. for $i \in \{x, y\}$ and every $\lambda \in U$, $\mathbb{E}_{X,Y}\left|(\partial_i \log p_\lambda(X,Y))^3\right| < \infty$.

2. $A_1^2 + A_2^2 + A_3^2 \neq 0$.

**Lemma 13.** *Under assumption 4, $A_i$ ($i = 0, 1, 2, 3$) are all well-defined integral and*

$$
\mathbb{E}_{X,Y}\left(\partial_x \log p_\lambda(X,Y)\right)^3 = A_3\lambda^3 + A_2\lambda^2 + A_1\lambda + A_0.
$$

**Proof**

$$
\mathbb{E}_{X,Y}\left(\partial_x \log p_\lambda(X,Y)\right)^3 = \int_{\mathbb{R}^2} \left(\partial_x \log p_\lambda(x,y)\right)^3 p_\lambda(x,y)dxdy
$$

$$
= \int_{\mathbb{R}^2} \frac{\left(\partial_x p_\lambda(x,y)\right)^3}{p_\lambda(x,y)^2} dxdy
$$

$$
= \int_{\mathbb{R}^2} \frac{\left(\partial_x h(x,v) - \lambda f'(x)\partial_v h(x,v)\right)^3}{h(x,v)^2} dxdv
$$

,

Consequently we can write the integrand as:

$$
\frac{\left(\partial_x h(x,v) - \lambda f'(x)\partial_v h(x,v)\right)^3}{h(x,v)^2} = \sum_{i=0}^{3} g_i(x,v)\lambda^i,
$$

where

$$
\begin{cases}
g_3(x,v) &= -\frac{f'(x)^3 \partial_v h(x,v)^3}{h(x,v)^2}, \\
g_2(x,v) &= 3\frac{f'(x)^2 \partial_v h(x,v)^2 \partial_x h(x,v)}{h(x,v)^2}, \\
g_1(x,v) &= -3\frac{f'(x)\partial_v h(x,v)\partial_x h(x,v)^2}{h(x,v)^2}, \\
g_0(x,v) &= \frac{\partial_x h(x,v)^3}{h(x,v)^2}.
\end{cases}
$$

As $h(x,v) > 0$, $g_i$ are continuous. Taking $\lambda_k \in U$ ($0 \le k \le 3$) s.t. $\lambda_i \neq \lambda_j$ for $i \neq j$. By assumption 4, for $0 \le k \le 3$, $\int_{\mathbb{R}^2}\left|\sum_{i=0}^{3} g_i(x,v)\lambda_k^i\right|dxdv < \infty$, i.e. $\sum_{i=0}^{3}\lambda_k^i g_i \in L^1(\mathbb{R}^2)$. Notice that the matrix $A = (A_{ik})_{0 \le i,k \le 3}$ such that $A_{ik} = \lambda_i^k$ is a Vandermonde matrix and thus invertible. Then since $L^1(\mathbb{R}^2)$ is a linear space, we conclude that $g_i \in L^1(\mathbb{R}^2)$ for $0 \le i \le 3$. Consequently $A_i = \int_{\mathbb{R}^2} g(x,v)dxdv$ is well-defined and

$$
\mathbb{E}_{X,Y}\left(\partial_x \log p(X,Y)\right)^3 = A_3\lambda^3 + A_2\lambda^2 + A_1\lambda + A_0.
$$

$\blacksquare$

**Proof of Proposition 3**

$$
p(x,y,z) = p(z)p(x|z)p(y|x,z) = p(z)q_0(x - \phi_0(z))\sigma(x)^{-1}q_1\left(\frac{y - \lambda f(x) - \phi_1(z)}{\sigma(x)}\right).
$$

We write $p_\lambda(x,y)$ as $p(x,y)$ for short, therefore,

$$p(x,y) = \int_{\mathbb{R}} p(z)q_0(x - \phi_0(z))\sigma(x)^{-1}q_1\left(\frac{y - \lambda f(x) - \phi_1(z)}{\sigma(x)}\right) dz.$$

Let $h(x,v) := \int_{\mathbb{R}} p(z)q_0(x - \phi_0(z))\sigma(x)^{-1}q_1\left(\frac{v - \phi_1(z)}{\sigma(x)}\right) dz$. Notice that $h$ is independent of $\lambda$, $f$ and $h \in C^{1,1}(\mathbb{R}^2)$ under our assumption. As $h(x, y - \lambda f(x)) = p(x,y)$, we derive:

$$\begin{cases} \partial_y p(x,y) &= \partial_v h(x, y - \lambda f(x)), \\ \partial_x p(x,y) &= \partial_x h(x, y - \lambda f(x)) - \lambda f'(x)\partial_v h(x, y - \lambda f(x)). \end{cases}$$

Our first observation is that: $\mathbb{E}_{X,Y}\left(\partial_y \log p(X,Y)\right)^3$ is independent of $\lambda$ and $f$, as is shown below:

$$\mathbb{E}_{X,Y}\left(\partial_y \log p(X,Y)\right)^3 = \int_{\mathbb{R}^2} \left(\partial_y \log p(x,y)\right)^3 p(x,y)dxdy$$

$$= \int_{\mathbb{R}^2} \frac{(\partial_y p(x,y))^3}{p(x,y)^2}dxdy$$

$$\text{(By change of variables)} = \int_{\mathbb{R}^2} \frac{(\partial_y p(x, v + \lambda f(x)))^3}{p(x, v + \lambda f(x))^2}dxdv$$

$$= \int_{\mathbb{R}^2} \frac{(\partial_v h(x,v))^3}{h(x,v)^2}dxdv.$$

On the other hand, by lemma 13 we get:

$$\left|\mathbb{E}_{X,Y}\left(\partial_x \log p(X,Y)\right)^3\right| = \left|A_3\lambda^3 + A_2\lambda^2 + A_1\lambda + A_0\right|.$$

Consequently, by assumption 4 $\left|\mathbb{E}_{X,Y}\left(\partial_x \log p(X,Y)\right)^3\right| \to +\infty$ as $\lambda \to +\infty$, thus there exists a constant $\lambda^*$ s.t. for every $\lambda \geq \lambda^*$:

$$\left|\mathbb{E}_{X,Y}\left(\partial_x \log p(X,Y)\right)^3\right| > \left|\mathbb{E}_{X,Y}\left(\partial_y \log p(X,Y)\right)^3\right|.$$

$$\blacksquare$$

**Remark 14.** *In particular, if $A_3 \neq 0$, let $R = \left|\mathbb{E}_{X,Y}\left(\partial_y \log p(X,Y)\right)^3\right|$, we can set $\lambda^* = 1 + \max\left\{\frac{3|A_2|}{|A_3|}, \sqrt{\frac{3|A_1|}{|A_3|}}, \sqrt[3]{\frac{3|A_0|+R}{|A_3|}}\right\}$. It is easy to verify that*

$$|A_3|\lambda^3 - |A_2|\lambda^2 - |A_1|\lambda - |A_0| > R,$$

*when $\lambda \geq \lambda^*$.*

## I    EXPLICIT COMPUTATION OF EXAMPLE 4

$$p(x,y,z) \sim \exp\left(-\frac{z^2}{2} - \frac{(x-z)^2}{2} - \frac{(y-z-f(x)^2)}{2}\right),$$

$$p(x,y) = \int_{\mathbb{R}} p(x,y,z)dz = C\exp\left(-\frac{2}{3}\left(x^2 + (y - f(x))^2 - x(y - f(x))\right)\right)$$

$$= C\exp\left(-\frac{2}{3}\left(y - \frac{x}{2} - f(x)\right)^2 - \frac{x^2}{2}\right),$$

$C = \frac{1}{\sqrt{3}\pi}$ is a normalization constant. Therefore,

$$\begin{cases} \partial_y \log p(x,y) &= -\frac{4}{3}\left(y - \frac{x}{2} - f(x)\right), \\ \partial_x \log p(x,y) &= \frac{4}{3}\left(y - \frac{x}{2} - f(x)\right)\left(\frac{1}{2} + f'(x)\right) - x \\ &= \frac{2}{3}\left((1 + 2f'(x))y - f(x) - 2x - xf'(x) - 2f(x)f'(x)\right) \\ &= \frac{2}{3}\left(\alpha(x)y - \beta(x)\right), \end{cases}$$

where $\alpha(x) = 1 + 2f'(x)$ and $\beta(x) = f(x) + 2x + xf'(x) + 2f(x)f'(x)$.

$$Skew_y = \frac{64C}{27} \left| \int_{\mathbb{R}} \int_{\mathbb{R}} (y - \frac{x}{2} - f(x))^3 e^{-\frac{2}{3}(y-\frac{x}{2}-f(x))^2} \mathrm{d}y \, e^{-\frac{x^2}{2}} \mathrm{d}x \right|$$

$$= \frac{64C}{27} \left| \int_{\mathbb{R}} \int_{\mathbb{R}} u^3 e^{-\frac{2}{3}u^2} \mathrm{d}u \, e^{-\frac{x^2}{2}} \mathrm{d}x \right| = 0.$$

On the other hand,

$$Skew_x = \frac{8C}{27} \left| \int_{\mathbb{R}} \int_{\mathbb{R}} (\alpha(x)y - \beta(x))^3 \, e^{-\frac{2}{3}(y-\frac{x}{2}-f(x))^2} \mathrm{d}y \, e^{-\frac{x^2}{2}} \mathrm{d}x \right|.$$

Let $\gamma(x) = \frac{x}{2} + f(x)$. Notice that $\alpha(x)\gamma(x) - \beta(x) = -\frac{3x}{2}$

$$\int_{\mathbb{R}} (\alpha(x)y - \beta(x))^3 \, e^{-\frac{2}{3}(y-\frac{x}{2}-f(x))^2} \mathrm{d}y = \int_{\mathbb{R}} (\alpha(x)(y + \gamma(x)) - \beta(x))^3 \, e^{-\frac{2}{3}y^2} \mathrm{d}y$$

$$= 3\alpha(x)^2(\alpha(x)\gamma(x) - \beta(x)) \int_{\mathbb{R}} y^2 e^{-\frac{2}{3}y^2} \mathrm{d}y + (\alpha(x)\gamma(x) - \beta(x))^3 \int_{\mathbb{R}} e^{-\frac{2}{3}y^2} \mathrm{d}y$$

$$= -\frac{9}{2}x\alpha(x)^2 \int_{\mathbb{R}} y^2 e^{-\frac{2}{3}y^2} \mathrm{d}y - \frac{27}{8}x^3 \int_{\mathbb{R}} e^{-\frac{2}{3}y^2} \mathrm{d}y.$$

Let $\mu = \int_{\mathbb{R}} y^2 e^{-\frac{2}{3}y^2} \mathrm{d}y = \frac{3\sqrt{6\pi}}{8}$, $\int_{\mathbb{R}} e^{-\frac{2}{3}y^2} \mathrm{d}y$.

$$Skew_x = \frac{8C}{27} \left| \int_{\mathbb{R}} \left( -\frac{9\mu}{2}x\alpha(x)^2 - \frac{27\mu_0}{8}x^3 \right) e^{-\frac{x^2}{2}} \mathrm{d}x \right|$$

$$= \frac{4C\mu}{3} \left| \int_{\mathbb{R}} x\alpha(x)^2 e^{-\frac{x^2}{2}} \mathrm{d}x \right|$$

$$= \frac{4C\mu}{3} \left| \int_{\mathbb{R}} x(1 + 2f'(x))^2 e^{-\frac{x^2}{2}} \mathrm{d}x \right|$$

$$= \frac{16C\mu}{3} \left| \int_{\mathbb{R}} x(f'(x) + f'(x)^2) e^{-\frac{x^2}{2}} \mathrm{d}x \right|$$

$$= \frac{2\sqrt{2\pi}}{\pi} \left| \int_{\mathbb{R}} x(f'(x) + f'(x)^2) e^{-\frac{x^2}{2}} \mathrm{d}x \right|.$$

## J   ADDITIONAL EXPERIMENTS

**Runtime** Table 1 presents the runtime, in seconds, for `SkewScore` using Sliced Score Matching (SSM) (Song et al., 2020), for various variable counts $d = \{10, 20, 50\}$, where the number of edges is twice the number of variables. All experiments are on 12 CPU cores with 24 GB RAM.

Table 1: Wall-clock runtime (in seconds) of `SkewScore` on ER2 graph with 1000 samples.

| Method | $d = 10$ | $d = 20$ | $d = 50$ |
|---|---|---|---|
| `SkewScore` (SSM) | $310.24 \pm 7.17$s | $704 \pm 19.82$s | $1825.92 \pm 39.23$s |

**Effect of Non-Symmetric Noise** We conducted experiments adhering to the same bivariate HSNM data generation process outlined in Section 6, except the noise now follows a Gumbel distribution. Our method remains relatively robust to Gumbel noise, a skewed distribution. The results in the table 2 demonstrate that the reduction in accuracy for SkewScore is less significant than for HOST, which operates under the assumption of Gaussian noise. LOCI and HECI are designed to handle various noise types, which unsurprisingly results in strong performance. However, these two methods are designed for bivariate models.

Table 2: Accuracies of estimated causal relationships in the bivariate case with a non-symmetric Gumbel noise. The results are averaged over 100 independent runs.

| Methods | HNMs (Gumbel) |
|---|---|
| SkewScore | 89 |
| LOCI | **100** |
| HECI | 99 |
| HOST | 33 |
| DiffAN | 92 |
| DAGMA | 27 |
| NoTears | 70 |

**Performance on Additive Noise Model (ANM)** Though `SkewScore` is designed for heteroscedastic noise, it is still valid on nonlinear models with homoscedastic noise. We explore the algorithm's performance using data generated according to Additive Noise Models (ANM) with a GP as the nonlinear function and Gaussian noise. After obtaining the topological order with `SkewScore`, CAM pruning (Bühlmann et al., 2014) is applied. We calculate the topological order divergence and the structural Hamming distance (SHD) across varying dimensions, focusing on graphs with $d$ and $2d$ edges, represented as ER1 and ER2 graphs, respectively. The results, presented in Tables 3 and 4, indicate that `SkewScore` maintains adequate performance in ANM settings. Results are averaged over 10 independent runs.

Table 3: Topological order divergence across dimensionalities ($d \in \{10, 20, 50\}$) using data generated from ANMs.

| | ER1 | | | ER2 | | |
|---|---|---|---|---|---|---|
| Methods | d=10 | d=20 | d=50 | d=10 | d=20 | d=50 |
| SkewScore | $\mathbf{1.00 \pm 1.00}$ | $\mathbf{3.10 \pm 0.70}$ | $\mathbf{8.50 \pm 3.17}$ | $3.80 \pm 2.44$ | $11.00 \pm 4.12$ | $\mathbf{23.40 \pm 5.82}$ |
| HOST | $3.90 \pm 1.45$ | $8.40 \pm 2.11$ | $15.00 \pm 3.74$ | $11.70 \pm 2.45$ | $22.20 \pm 3.79$ | $51.60 \pm 3.77$ |
| DiffAN | $1.80 \pm 0.98$ | $4.50 \pm 3.07$ | $14.89 \pm 2.18$ | $\mathbf{2.70 \pm 1.19}$ | $\mathbf{7.00 \pm 2.32}$ | $23.43 \pm 2.50$ |
| DAGMAMLP | $3.70 \pm 1.10$ | $7.70 \pm 1.79$ | $15.70 \pm 4.65$ | $10.20 \pm 0.75$ | $17.10 \pm 1.51$ | $35.00 \pm 2.37$ |

**Effect of Sample Size** Table 5 illustrates the order divergence for various models on synthetic data generated, measured across increasing sample sizes $n = \{100, 1000, 10000\}$. This analysis was conducted with a fixed dimension $d = 10$ with an equal number of edges, utilizing an ER graph model. The results indicate that our method, specifically `SkewScore`, demonstrates enhanced performance as the sample size increases. This trend suggests that `SkewScore` can efficiently leverage larger

Table 4: SHD across different dimensionalities ($d \in \{10, 20, 50\}$) using data generated from ANMs.

| Methods | ER1 | | | ER2 | | |
|---|---|---|---|---|---|---|
| | d=10 | d=20 | d=50 | d=10 | d=20 | d=50 |
| SkewScore | **2.90 ± 1.47** | **7.08 ± 2.32** | **19.32 ± 7.26** | 8.26 ± 2.42 | 30.50 ± 5.52 | **76.50 ± 5.50** |
| HOST | 7.20 ± 2.04 | 17.30 ± 3.95 | 44.70 ± 6.56 | 17.80 ± 2.44 | 37.10 ± 4.04 | 95.70 ± 6.42 |
| DiffAN | 3.50 ± 1.40 | 9.90 ± 1.87 | 23.56 ± 1.27 | **7.31 ± 1.42** | **30.10 ± 4.06** | 82.29 ± 5.95 |
| DAGMAMLP | 6.90 ± 1.04 | 15.60 ± 1.28 | 37.60 ± 4.18 | 12.30 ± 0.64 | 37.50 ± 1.20 | 92.40 ± 1.62 |

datasets to improve estimation accuracy. Notably, even with smaller sample sizes, `SkewScore` maintains commendable performance, underscoring its robustness and effectiveness in scenarios with limited data availability. This capability makes it particularly advantageous in practical applications where acquiring large volumes of data may be challenging.

Table 5: Topological order divergence on synthetic data is analyzed as a function of sample size. We fix $d = 10$ with the same number of edges, using an ER graph model, and vary the sample size across $n = \{100, 1000, 10000\}$. Results are provided for both the HSNM and the HSNM with latent confounders.

| | HSNMs | | | HSNMs with Confounders | | |
|---|---|---|---|---|---|---|
| | n=100 | n=1000 | n=10000 | n=100 | n=1000 | n=10000 |
| SkewScore | **1.90 ± 1.70** | **0.90 ± 0.75** | **0.60 ± 0.32** | **2.10 ± 0.83** | **1.30 ± 0.64** | **1.10 ± 0.58** |
| HOST | 4.40 ± 1.91 | 4.40 ± 1.11 | 3.80 ± 1.47 | 4.56 ± 1.34 | 4.20 ± 1.60 | 3.90 ± 1.45 |
| DiffAN | 4.70 ± 1.68 | 2.30 ± 1.19 | 2.40 ± 0.92 | 3.60 ± 1.50 | 2.80 ± 1.47 | 2.75 ± 1.39 |
| VarSort (Reisach et al., 2021) | 2.10 ± 1.45 | 2.00 ± 1.38 | 1.50 ± 1.81 | 2.20 ± 1.54 | 2.10 ± 1.51 | 1.90 ± 1.37 |

**Real Data** Our method's accuracy is 62.6% in the Tübingen cause-effect pairs dataset, excluding multivariates and binary pairs (47, 52-55, 70, 71, 105 and 107) as Immer et al. (2023). According to the reported results from Immer et al. (2023), our method performance is slightly better than LOCI. GRCI Strobl & Lasko (2023) performs the best, and QCCD Tagasovska et al. (2020) also performs quite well. These methods are likely more robust to assumption violation in real data such as low sample complexity. The Tübingen cause-effect pairs dataset contains discrete datasets, where the score function is not well-defined, and datasets that do not satisfy the symmetric heteroscedastic assumption, which is challenging for our method.

**Multivariate extension of the case study in Section 5** Finally, we conduct experiments to explore the multivariate extension of the case study in Section 5. The data generation process for experiments for multivariate HSNMs with latent confounders is based on Montagna et al. (2023a). Let $Z \in \mathbb{R}^d$ represent the latent common cause. For each pair of distinct nodes $X_i$ and $X_j$, a Bernoulli random variable $C_{ij} \sim \text{Bernoulli}(\rho)$ is sampled, where $C_{ij} = 1$ implies a confounding effect $Z$ between $X_i$ and $X_j$. The parameter $\rho$ determines the sparsity of confounded pairs in the graph. In our experiments, we choose $\rho = 0.2$. Data generation follows a HSNM with latent confounders: $X_i = f_i(\text{pa}_X(X_i)) + \phi_i(\text{pa}_Z(X_i)) + \sigma_i(\text{pa}_X(X_i)) \cdot N_i$, for each variable $i = 1, \ldots, d$. Here, $\text{pa}_X(X_i)$ and $\text{pa}_Z(X_i)$ represent the observed and hidden variable parents of $X_i$, respectively. The model incorporates the Gaussian processes with an RBF kernel of unit bandwidth to generate the function $f_i$, a linear function $\phi_i$, and a conditional standard deviation $\sigma_i$ modeled as a sigmoid function of the observed parents' linear combination. For a given number of observed variables $d$, we adjust the sparsity of the sampled graph by setting the average number of edges to either $d$ (ER1) or $2d$ (ER2). The results are shown in 6 and 7, where the lower topological order divergence means better performance. In general, `SkewScore`, Diffan, and NoTears have the best performances. In sparser graph (ER1), `SkewScore` performs the best when $d \in \{10, 20, 30, 40\}$. In the ER2 graph, `SkewScore` and Diffan have similar good performances. The theoretical insights for this extension of the case study in Section 5 will be the focus of future work.

The data used for the barplots in Figures 3a and 3b are provided in Table 8. `SkewScore` maintains at least 95% accuracy across all eight settings in Table 8, while other baselines fall below 85% accuracy in at least one setting.

Table 6: Topological order divergence: Synthetic data generated using multivariate HSNM with latent confounders with an ER1 graph.

| Method | d=10 | d=20 | d=30 | d=40 | d=50 |
|---|---|---|---|---|---|
| SkewScore | $\mathbf{1.20 \pm 0.98}$ | $\mathbf{2.40 \pm 2.33}$ | $\mathbf{4.60 \pm 1.80}$ | $\mathbf{6.60 \pm 3.29}$ | $7.20 \pm 2.64$ |
| HOST | $4.00 \pm 1.48$ | $8.10 \pm 1.64$ | $10.40 \pm 2.29$ | $14.90 \pm 2.66$ | $17.50 \pm 2.73$ |
| DiffAN | $2.80 \pm 1.17$ | $5.60 \pm 2.62$ | $8.60 \pm 2.91$ | $9.50 \pm 2.46$ | $11.60 \pm 3.67$ |
| DAGMAMLP | $6.00 \pm 2.00$ | $14.50 \pm 2.11$ | $19.30 \pm 2.49$ | $27.10 \pm 3.33$ | $34.50 \pm 4.13$ |
| NoTears | $1.50 \pm 1.02$ | $5.30 \pm 2.45$ | $5.00 \pm 2.57$ | $6.80 \pm 3.97$ | $\mathbf{6.30 \pm 2.45}$ |

Table 7: Topological order divergence: Synthetic data generated using multivariate HSNM with latent confounders with an ER2 graph.

| Method | d=10 | d=20 | d=30 | d=40 | d=50 |
|---|---|---|---|---|---|
| SkewScore | $\mathbf{3.00 \pm 1.61}$ | $8.60 \pm 2.11$ | $14.00 \pm 4.73$ | $\mathbf{16.60 \pm 2.42}$ | $25.80 \pm 4.26$ |
| HOST | $11.80 \pm 2.44$ | $22.80 \pm 4.45$ | $32.90 \pm 5.47$ | $41.90 \pm 6.55$ | $51.80 \pm 6.27$ |
| DiffAN | $3.30 \pm 2.10$ | $\mathbf{7.20 \pm 2.40}$ | $\mathbf{13.20 \pm 3.49}$ | $18.10 \pm 3.86$ | $\mathbf{22.10 \pm 4.44}$ |
| DAGMAMLP | $17.60 \pm 1.80$ | $34.90 \pm 2.47$ | $54.20 \pm 2.18$ | $74.40 \pm 2.15$ | $91.20 \pm 3.19$ |
| NoTears | $8.10 \pm 2.81$ | $16.70 \pm 6.81$ | $20.90 \pm 5.73$ | $25.30 \pm 7.06$ | $27.00 \pm 4.45$ |

Table 8: Accuracy (%) of causal direction estimation across different data generation processes for bivariate and latent-confounded triangular heteroscedastic noise models. These data were used for the barplots in Figures 3a and 3b. Additional notes: **Dark green bold** indicates the best performance, and dark green regular indicates the second best performance.

| | Bivariate (no latent confounders) | | | | Latent-confounded Triangular | | | |
|---|---|---|---|---|---|---|---|---|
| | GP-Sig (Gauss) | GP-Sig (t) | Sig-abs (Gauss) | Sig-abs (t) | GP-Sig (Gauss) | GP-Sig (t) | Sig-abs (Gauss) | Sig-abs (t) |
| SkewScore | 99 | **100** | 98 | 95 | **98** | **97** | 96 | **100** |
| LOCI | 99 | 99 | 98 | **96** | 68 | 83 | 99 | **100** |
| HECI | **100** | 95 | **100** | 64 | 54 | 64 | 99 | 83 |
| HOST | **100** | 81 | **100** | 89 | 89 | 91 | **100** | 91 |
| DiffAN | 92 | 89 | 92 | 48 | 76 | 90 | 80 | 35 |
| DAGMA | 72 | 58 | 58 | 26 | 45 | 61 | 5 | 40 |
| NoTears | 85 | 90 | **100** | 48 | 55 | 55 | 13 | 52 |

