# OpenReview forum: "A Skewness-Based Criterion for Addressing Heteroscedastic Noise in Causal Discovery"
_ICLR.cc/2025/Conference — ICLR 2025 Poster_

### Official Review · Reviewer_iLTQ · 2024-11-03

**Soundness:** 3
**Presentation:** 4
**Contribution:** 3
**Rating:** 6
**Confidence:** 3

**Summary:**

In this paper, the author proposed a novel causal learning algorithm for heteroscedastic symmetric noise models (HSNMs), where the noise term of a variable $X_i$ can be parameterized by $\sigma(pa(X_i))N_{X_i}$ with symmetric noise $N_{X_i}$. Specifically, the author showed that the skewness of the score function (SkewScore) in the $X_i$-coordinate is zero if and only if $X_i$ is a leaf node under certain assumptions. Therefore, the topological order can be identified by iteratively removing the variables with the least SkewScore. Conditional independence tests are then applied to determine the edges. Its validity for the HSNM with latent confounder is also discussed. Extensive experiments showed that the proposed method is effective on HSNMs and is robust to assumption violations.

**Strengths:**

1. The method is novel and can be potentially applied to causal discovery with latent confounders.
2. The theoretical justification of Assumption 1 seems sound, and the experiments showed that the proposed method works particularly well on sparse HSNMs.
3. The author extensively evaluated the method when the assumptions are violated, e.g. non-symmetric noise distribution, and latent confounder in the multivariate setting, and proved its effectiveness and robustness.

**Weaknesses:**

1. The author mainly used ER1 to evaluate the methods on HSNMs without latent confounders, which is relatively sparse.
2. I think the justification of Equation (6) was a bit missing. Proposition 5 justified that Assumption 1 is mild, and therefore Equation (5) in the bivariate setting holds. However, there is no such explanation for Equation (6). Can you provide some concrete examples where Equation (6) is reduced to some known results, or simplified by specific function and noise classes?

**Questions:**

1. Which ER graph model is used in the experiments, $G(d, p)$ or $G(d, m)$?
2. Can you explain in more detail why $c\neq0$ in general in Remark 6?
3. What is the runtime for the order learning and independent tests separately?

---

> ### Author Response · Authors · 2024-11-22
>
> > The author mainly used ER1 to evaluate the methods on HSNMs without latent confounders, which is relatively sparse.
>
> Thank you for your feedback. We have conducted experiments on ER2 graphs without latent confounders. The results are presented in the table below, where SkewScore consistently outperforms the other methods.
>
> | Method| $d=10$| $d=20$| $d=50$|
> |----|----------|----------|-----------------|
> | SkewSCORE  | $2.70 \pm 1.55$ | $7.80 \pm 3.25$ | $25.50 \pm 2.54$ |
> | HOST       | $12.40 \pm 2.20$| $21.00 \pm 3.69$| $54.30 \pm 6.23$ |
> | DAGMAMLP   | $17.60 \pm 1.43$| $36.10 \pm 2.47$| $90.80 \pm 2.79$ |
> | NoTEARS    | $9.90 \pm 2.34$ | $20.00 \pm 5.55$| $36.50 \pm 5.73$ |
>
> Table 1: Topological order divergence with different Dimensionalities: ER2
>
> > I think the justification of Equation (6) was a bit missing. Proposition 5 justified that Assumption 1 is mild, and therefore Equation (5) in the bivariate setting holds. However, there is no such explanation for Equation (6). Can you provide some concrete examples where Equation (6) is reduced to some known results, or simplified by specific function and noise classes?
>
> Thank you for your question. The multivariate extension of Proposition 5 is natural, and we provided the discussion in Lines 778-787 of the manuscript. To enhance clarity, we revise the previous discussion and provide the following explanation:
>
> Similar to Proposition 5, the set of functions $ (f_i,\sigma_i) $ that violate Eq. (6) has a property analogous to having Lebesgue zero measure. Specifically, we parameterize the causal model by parameterizing functions $ f_i^\theta $ and $ \sigma_i^\theta $, where $ \theta $ belongs to the parameter space $ \Theta \subset \mathbb{R}^K $.
> The left-hand side of Eq. (6) defines a function of the parameter $ \theta $, and we write it as $ F(\theta) $.
> With this parameterization, the model that violates Eq. (6) corresponds to the solution of the equation $ F(\theta)=0 $. Under regular assumptions on $ f_i $, $ \sigma_i $, $ p_X $, and $ p_{N_i} $, $ F $ is a real-analytic function. Then by applying (Mityagin, 2015, Proposition 0), the violated set $ S_{\text{nc}}(\Theta) = \\{\theta \in \Theta : F(\theta) = 0\\} $ has zero measure as long as the function $ F $ is not a zero constant function. It is worth noting, however, that in the homoscedastic Gaussian linear model, $ F $ is identically zero, and thus (Mityagin, 2015, Proposition 0) no longer applies. We have included this discussion in Lines 857-866 in the revision. Thanks again for your suggestion.
>
>
> We provide an explicit example involving 3 nodes. The model is given by:
> $X = f_ 2(Z) + N_ 1$, $ Y = f_ 1(Z) + f_ 3(X) + N_ 2$, where $ Z,N_ 1, N_ 2 $ are independent Gaussian variables with distribution $\mathcal{N}(0,1)$, and $f_ i(x) = a_ ix^2+b_ ix+c_ i$ are quadratic functions. In this case, Equation (6) can be simplified into: $a_ 2 b_ 2 + a_ 3 b_ 3\neq 0$, $a_ 3\neq 0$, which are non-restrictive and can be easily satisfied.
>
> > Which ER graph model is used in the experiments, $G(d, p)$ or $G(d, m)$?
>
> $G(d, m)$ is used. Thanks for the question.
>
> > Can you explain in more detail why $c \neq 0$ in general in Remark 6?
>
> Thank you for your question.  As discussed in the first response to Reviewer 8vJA, a non-constant $\sigma(x)$ will make the summation term in Equation (*) more complicated and "non-degenerated". Intuitively, the complicated interactions between the link functions in the summation term are less likely to cancel out unless the model is designed artificially.
> Specifically, when $p_ X$ and $\sigma(x)$ are both symmetric function and their symmetric axis-es coincide, then $c=0$.
>
> More rigorously, if we parameterize $\sigma$ as $\sigma(\theta)$, where $\theta$ belongs to the parameter space $\Theta$. Then similar to the proof of proposition 5, under smooth assumption, we could prove that $\\{\theta:c(\theta) = 0\\}$ has 0 Lebesgue measure. This illustrates that $c(\theta)\neq 0$ generally.
>
> >  What is the runtime for the order learning and independent tests separately?
>
> Thank you for your question. The runtime for order learning and independent tests separately is provided below. As shown in Table 2, the runtime for both order learning and DAG recovery increases with the number of variables $d$. However, the rate of increase for DAG recovery is higher compared to order learning. This suggests that while both phases are affected by dimensionality, DAG recovery tends to become more computationally expensive in higher-dimensional settings.
>
>
>
> | Phase | $d=10$ | $d=20$ | $d=50$  |
> |--------------------|-----------------------|----------------------|-----------------------|
> | Order learning     | $308.16 \pm 8.20$ s   | $712 \pm 17.62$ s| $1794.23 \pm 41.56$ s |
> | DAG recovery (KCI) | $62.87 \pm 2.21$ s    | $324.04 \pm 15.42$ s | $2531.14 \pm 46.72$ s |
>
> Table 2: Wall-clock runtime (in seconds) of SkewScore with 1000 samples. The experiment is run on 12 CPU cores with 24 GB RAM.

---

> ### Author Response · Authors · 2024-12-01
> **Looking forward to discussion**
>
> Dear Reviewer iLTQ,
>
> Thank you for your time and effort in reviewing our paper. We have carefully addressed your concerns in the rebuttal and would greatly appreciate it if you could review our response. If you have further questions or comments after reading the rebuttal, we hope to have the opportunity to address them.
>
> Thanks again for your time and consideration.
>
> Best regards,
> Authors of Paper 9319

---

### Official Review · Reviewer_8vJA · 2024-11-03

**Soundness:** 3
**Presentation:** 4
**Contribution:** 4
**Rating:** 10
**Confidence:** 4

**Summary:**

==Post-rebuttal update==

The authors' replies to my questions show their further insights into the problem and how the work could be deepened.

I also had a rough scan of other reviews and their respective replies, and I hold my original score.

This is an exceptionally well-written paper with valuable insights and high potential impact. Of course, no papers are perfect, but the current rating best describes my evaluation of the paper, i.e., it "should be highlighted at the conference".

==End update==

This paper proposes a method to identify causal directions in heteroscedastic symmetric noise models (HSNMs) using the skewness of the score function, which is the gradient of the log density. The skewness considered here is analogous to standard skewness but is distinct, offering computational advantages. Importantly, the paper proves that the score's skewness is zero only in the causal direction. This idea is later generalized to the multivariate case. The proposed method also shows robustness in the presence of hidden confounders, and a theoretical analysis is provided for an "additive" confounder in the bivariate case. The method demonstrates strong empirical performance.

**Strengths:**

The paper provides a neat and effective solution to a critical problem. The connection between score skewness and causal direction is both theoretically grounded and practically effective.

The extensions to multivariate cases and the potential applicability under hidden confounding are valuable contributions.

The theoretical analysis appears rigorous (though I haven’t verified every detail).

The paper is clearly written, with helpful elements like the two properties of the score function, the motivation of score skewness by analogy to standard skewness, and well-designed figures.

**Weaknesses:**

The generality of Assumption 1, which defines the applicable HSNMs, could be explored further, although the paper presents sufficient results for a conference submission.

The method's applicability under hidden confounding might be limited to special confounding structures.

**Questions:**

### On Proposition 5
Could you elaborate on why, “when $\sigma(x)$ is not constant (i.e., in the heteroscedastic model), $c \neq 0$ in most cases”? More broadly, could you comment on the condition $c=0$? This characterization could be crucial as it defines a class of distributions $p(u, x)$.

Proposition 5 considers $f$ in a finite-dimensional linear space, specifically polynomials. Could neural networks with hidden layers be considered also (since Example 7 doesn’t)?

It’s interesting that $f$ is required to lie in a “large finite-dimensional space.” I would assume that if $f$ were in an infinite-dimensional functional space, the space should instead be “small.” Could you share your thoughts on this apparent contrast?

### Additional Comments
Could similar theoretical results be proven for standard skewness?

Assumption 4: Typically, “regularity” refers to the properties of a well-behaved functional space. In Assumption 4, condition 1) is indeed a regularity condition, but it’s unclear if condition 2) qualifies as such. Additionally, could you explain why condition 2) is considered mild?

Minor: There is a duplicate bibliography entry for Mityagin (2015).

---

> ### Author Response · Authors · 2024-11-22
>
> > The generality of Assumption 1, which defines the applicable HSNMs, could be explored further, although the paper presents sufficient results for a conference submission.
>
> Thank you for your suggestion. We further discuss the generality of Assumption 1 as follows.
> The score function of HSNMs given by Eq. (1) in the manuscript is given by (provided Eq. (14), Line 1063)
> $$
>     \partial_ {x_ k}\log p(x) = \frac{p'_ k}{\sigma_ kp_ k}-\sum_ {i\in child(k)}\left( \frac{p'_ i}{p_ i}\cdot\frac{\sigma_ i\partial_ {x_ k}f_ i+(x_ i-f_ i)\partial_ {x_ k}\sigma_ i}{\sigma_ i^2}+\frac{\partial_ {x_ k}\sigma_ i}{\sigma_ i}\right),  (\*)
>     $$
> for $k \in \{ 1, ... , d\}$. If $k$ is a leaf node (sink node), it has no children, thus reducing the above equation to $\partial_ {x_ k}\log p(x) = \frac{p'_ k}{\sigma_ kp_ k}$. Roughly speaking, we aim to rely on this reduction to identify the leaf node, and therefore restrict the summation term in Equation (*) to be "non-degenerated", meaning the complicated interactions between the link functions in this summing term do not cancel out, thus making the score skewed. Assumption 1 imposes a formal restriction to prevent such degeneration in terms of skewness.
>
> We provide the following key intuition on Proposition 5: if we parameterize the link functions $f$ with parameter $\theta$, then Assumption 1 reduces to the inequality $F(\theta)\neq 0$ (for some analytic function $F$), which can generally hold when the $F$ isn't constant zero.
>
> Specifically, we first parameterize the HSNM by assuming the structures of the link functions $f$, e.g., using polynomials. Then, the left-hand side of the equation in Assumption 1 defines a function of the parameter $\theta$, and we write it as $F(\theta)$. With this parameterization, the model that violates Assumption 1 corresponds to the solution of the equation $F(\theta)=0$. Under regular assumptions on $ f_ i $, $ \sigma_ i $, $ p_ X $, and $ p_ {N_ i} $, $ F $ is a real-analytic function. Then by applying [Mityagin 2015, Proposition 0], the solution set $ S_ {\text{nc}}(\Theta) = \\{\theta \in \Theta : F(\theta) = 0\\} $ has zero measure as long as the function $F$ is not a constant zero function. Therefore, it suffices to verify whether $F$ is a zero constant function. When $F$ is a polynomial function, checking its non-constancy-zero is straightforward: verifying that at least one coefficient is non-zero ensures $F$ is not zero constant. We consider $f$ from a linear space, making $F$ a polynomial function of $\theta$. The constant term $c$ in Proposition 5 corresponds to the zero-power term of $F$. A non-zero $c$ guarantees $F$ is not constant zero. If $c = 0$, other coefficients of $F$ should be checked, as further detailed in Case 2 of Proposition 5.
>
> Thank you again for your suggestion; We have included the discussion in Lines 857-866 in the revision.
>
>
> > The method's applicability under hidden confounding might be limited to special confounding structures.
>
> Thank you for your feedback. You are correct that in our case study, we assume specific structures on the confounding effects. Extending the method to accommodate more general confounding structures is an important direction for future work. A key step in this extension would be relaxing the assumption of the additive structure. Addressing this would necessitate a new approach to quantify the impact of latent confounders on the link functions.
>
> > Could you elaborate on why, “when $\sigma(x)$ is not constant (i.e., in the heteroscedastic model), $c \neq 0$ in most cases”? More broadly, could you comment on the condition $c = 0$? This characterization could be crucial as it defines a class of distributions $p(u, x)$.
>
> Thanks for the question. Following the discussion in the first question, a non-constant $\sigma(x)$ will make the summation term in Equation (*) more complicated and "non-degenerated". Intuitively, the complicated interactions between the link functions in the summation term are less likely to cancel out unless the model is designed artificially.
> Specifically, when $p_ X$ and $\sigma(x)$ are both symmetric function and their symmetric axis-es coincide, then $c=0$.
>
> More rigorously, if we parameterize $\sigma$ as $\sigma(\theta)$, where $\theta$ belongs to the parameter space $\Theta$. Then similar to the proof of proposition 5, under smooth assumption, we could prove that $\\{\theta:c(\theta) = 0\\}$ has 0 Lebesgue measure. This illustrates that $c(\theta)\neq 0$ generally.
>
>
> For example, let $X,N\sim \mathcal{N}(0,1)$, $\sigma(x) = e^{ax+b}$. So the parameter is $\theta = (a,b)$ and $\Theta = \mathbb{R}^2$. We compute that
> $$
> c(a,b)= -a^3 - \frac{3a}{4} - \frac{3a}{4} e^{-b} e^{\frac{a^2}{2}} (1 + 3a^2) - \frac{9a^3}{4} e^{-2b} e^{2a^2} + \frac{15a^3}{8} e^{-3b} e^{\frac{9a^2}{2}}.
> $$
>
> The set $\\{(a,b)\in\mathbb{R}^2: c(a,b) = 0\\}$ is an analytic curve in $\mathbb{R}^2$, and thus has 0 measure.

---

> > ### Author Response · Authors · 2024-11-22
> >
> > > Proposition 5 considers $f$ in a finite-dimensional linear space, specifically polynomials. Could neural networks with hidden layers be considered also (since Example 7 doesn’t)?
> >
> > Yes, neural networks can also be considered. For example, if $f$ is a neural network with a real analytic activation function (e.g., Sigmoid, Softplus), similar results as Proposition 5 can hold. In Proposition 5, we consider $f$ in a linear space to simplify the computation and analysis.
> >
> > > It’s interesting that $f$ is required to lie in a “large finite-dimensional space.” I would assume that if $f$ were in an infinite-dimensional functional space, the space should instead be “small.” Could you share your thoughts on this apparent contrast?
> >
> > This is a very good question. We assume $f$ lies in a sufficiently large space to avoid it being restricted to overly simple spaces, which could lead to underfitting. Additionally, assuming $f$ lies in a sufficiently large space helps make the summation term in (*)  non-degenerated to encode asymmetry information.
> >
> > On the other hand, infinite-dimensional spaces come with the new challenge of finding a proper measure (and corresponding Borel algebra) on the infinite-dimensional space that is computationally feasible in our setting.
> >
> >
> > An example is the Wiener measure on the space of continuous functions, which is induced by Brownian motion. However, computations involving this measure are inherently more challenging, as the integral in Equation (6) provides global information about the function, whereas the Borel $\sigma$-algebra of the continuous function space is inherently tied to local properties. Additionally, the definition of the score requires the link functions to be differentiable. This suggests that the continuous function space might be too large for our problem, and we should consider smaller spaces, (e.g. the space of analytic functions), which impose additional restrictions on the functions. It is worth noting that defining an infinite-dimensional version of the Lebesgue measure on such spaces remains an open problem.
> >
> >
> >
> > Thanks again for this insightful question.
> >
> >
> > > Could similar theoretical results be proven for standard skewness?
> >
> > Thanks for the question. Yes, similar theoretical results for standard skewness can be proven. However, in heteroscedastic symmetric noise models, we need to estimate the skewness of the residuals or the conditional probability, which is challenging to compute in practice and often requires pre-extraction of the exogenous noise term. The SkewScore serves as a computationally tractable surrogate for the standard skewness to identify the causal ordering.
> >
> >
> > > Assumption 4: Typically, “regularity” refers to the properties of a well-behaved functional space. In Assumption 4, condition (1) is indeed a regularity condition, but it’s unclear if condition (2) qualifies as such. Additionally, could you explain why condition (2) is considered mild?
> >
> > Yes, the reviewer is correct, thanks for pointing this out. Assumption 4 (1) is a regularity assumption, while (2) isn't. We have corrected it in the revision.
> >
> > Assumption 4 (2) is similar to Assumption 1, but it is weaker, because it requires only one of $\{A_ 1,A_ 2,A_ 3\}$ to be non-zero. So similar to proposition 5, we could prove that $\{f:A_ 1=A_ 2=A_ 3 = 0\}$ has 0 measure. We should also point out that, for the homoscedastic Gaussian linear model (that is: we consider linear $f$ in example 4), we always have $A_ 1=A_ 2=A_ 3 = 0$, so the assumption fail to hold.
> >
> >
> > > There is a duplicate bibliography entry for Mityagin (2015).
> >
> > Thank you for pointing this out. We have corrected it in our latest revision.

---

> > > ### Comment · Reviewer_8vJA · 2024-11-28
> > > **Thank you for the rebuttal**
> > >
> > > The authors' replies to my questions show their further insights into the problem and how the work could be deepened.
> > >
> > > I also had a rough scan of other reviews and their respective replies, and I hold my original score.
> > >
> > > This is an exceptionally well-written paper with valuable insights and high potential impact. Of course, no papers are perfect, but the current rating best describes my evaluation of the paper, i.e., it "should be highlighted at the conference".

---

> > > > ### Author Response · Authors · 2024-11-29
> > > > **Thank you for your support**
> > > >
> > > > Thank you for your efforts and time in reviewing this paper. We greatly appreciate your support.

---

### Official Review · Reviewer_8P9r · 2024-11-04

**Soundness:** 3
**Presentation:** 3
**Contribution:** 2
**Rating:** 5
**Confidence:** 2

**Summary:**

The paper presents a novel approach to uncovering causal relationships in data with heteroscedastic noise, which challenges traditional causal discovery models. The authors propose the use of a skewness-based criterion derived from the score of data distributions to identify causal directions, establishing that this measurement is zero in the causal direction but non-zero in the anti-causal direction.

**Strengths:**

1. The introduction of the SkewScore, which is leveraged to identify causal structures without requiring the extraction of exogenous noise.
2. A theoretical extension of the criterion to multivariate cases, along with empirical validations showing its robustness, particularly in scenarios involving latent confounders.

**Weaknesses:**

See questions.

**Questions:**

1. How is Theorem 1 applied to establish the causal direction between variables $X$ and $Y$ in the experiments? Equations (4) and (5) represent the population mean, while in experiments, we can only obtain the sample mean.
2. The performance of SkewScore in terms of accuracy, as shown in Figure 3, is not consistently superior to other methods. Specifically, SkewScore outperforms others in the case of GP-sig Student's t in Figure 3(a), and in the cases of GP-sig Gaussian and GP-sig Student's t in Figure 3(b). What are the computation times associated with the different methods?

**Details Of Ethics Concerns:**

No.

---

> ### Author Response · Authors · 2024-11-21
>
> > How is Theorem 1 applied to establish the causal direction between variables $X$ and $Y$ in the experiments? Equations (4) and (5) represent the population mean, while in experiments, we can only obtain the sample mean.
>
> Thank you for your question. Theorem 1 is to establish identifiability results in the infinite sample limit, which serves as a foundational step towards establishing guarantees on finite samples. We acknowledge that in practice, we can only compute the sample mean as an estimator of the population mean. Under standard regularity conditions, the sample mean is a consistent estimator of the population mean; as the sample size increases, the sample mean converges to the true population mean. Therefore, we use the sample estimates of skewness as proxies for their population counterparts in applying Theorem 1.
>
> To assess the impact of using sample estimates, we conducted experiments examining the effect of sample size on our method’s performance (see Table 5 in Appendix J of our manuscript). The results show that our empirical skewness estimates become more accurate with larger sample sizes. Notably, even with smaller sample sizes, our method maintains strong performance, suggesting robustness to finite-sample variations.
>
> Our identifiability results in the infinite sample limit demonstrate that the causal direction can be identified under appropriate assumptions. Extending our theoretical findings to finite samples is an important avenue for future work, including extending the results of [Zhu et al., 2024] to non-Gaussian scenarios. Additionally, to further enhance robustness in finite samples, future work will explore hypothesis testing approaches for detecting non-zero skewness. Since we do not assume a specific noise distribution, non-parametric methods such as bootstrapping and permutation tests can be employed to assess the significance of our skewness estimates.
>
> In summary, although Theorem 1 is based on population means, we apply it in practice using sample estimates, and our experiments indicate that our method is effective even with finite samples.
>
>
> > The performance of SkewScore in terms of accuracy, as shown in Figure 3, is not consistently superior to other methods. Specifically, SkewScore outperforms others in the case of GP-sig Student's t in Figure 3(a), and in the cases of GP-sig Gaussian and GP-sig Student's t in Figure 3(b). What are the computation times associated with the different methods?
>
> Thank you for your feedback. We would like to emphasize that SkewScore consistently achieves at least 95\% accuracy across all eight settings in Figure 3, whereas other baselines drop below 85\% in at least one setting. This highlights the robustness of SkewScore, which we have clarified in the revised manuscript by including detailed numbers in Appendix J for completeness. Additionally, while SkewScore shows comparable or superior accuracy in the bivariate cases presented in Figure 3, it provides a significant advantage in more complex multivariate tasks, as demonstrated in Figure 4.
>
> Regarding computation times, we have summarized them in the following table for clarity. While SkewScore’s runtime is higher than HECI, it is still faster than LOCI, providing a balanced trade-off between efficiency and performance. SkewScore’s superior accuracy and robustness make it especially valuable for multivariate settings. Additionally, it is worth noting that methods like LOCI and HECI are specifically designed for two-variable scenarios, and extending them to multivariate cases presents considerable challenges.
>
> | Method  | Gaussian | (no latent)| Student's t | (no latent)| Gaussian| (latent) | Student's t | (latent)|
> |-----------|-------------|-------------|-------------|-------------|-------------|-------------|-------------|-------------|
> |           | Accuracy (%)    | Time (s)       | Accuracy (%)     | Time (s)        | Accuracy (%)     | Time (s)        | Accuracy (%)     | Time (s)         |
> |SkewScore|99|35.95±5.24|100|36.33±4.64|98|38.12±7.14|97|37.23±9.25|
> |LOCI|99|71.23±1.45|99|54.47±1.41|68|53.24±2.10|83|60.11±2.21|
> |HECI|100|0.08±0.03|95|0.07±0.02|54|0.13±0.02|64|0.12±0.01|
> |HOST|100|5.92±2.80|81|5.73±2.79|89|3.82±1.90|91|3.75±1.92|
> |DiffAN|92|44.00±9.13|89|45.33±9.20|76|48.33±9.72|90|46.65±9.46|
> |DAGMA|72|38.02±19.00|58|50.88±36.25|45|30.79±21.75|61|32.79±22.28|
> |NoTears|85|5.49±2.70|90|5.69±2.75|55|3.05±1.74|55|2.96±1.78|
>
> **Table**: Computation times (in seconds) for different methods in the pairwise settings presented in the paper. Results are shown for GP-Sig (Gaussian) and GP-Sig (Student's t), both with and without latent confounders. Experiments are run on 12 CPU cores with 24 GB RAM.
>
> References:
>
> Zhu, Z., Locatello, F., \& Cevher, V. (2024). Sample complexity bounds for score-matching: causal discovery and generative modeling. Advances in Neural Information Processing Systems, 36.

---

> > ### Comment · Reviewer_8P9r · 2024-11-21
> >
> > In the experiments, you can obtain a sample version of $SkewScore_y(p)$.How do you use the sample version of $SkewScore_y(p)$ to determine the causal direction? Do you determine the causal direction based on whether the sample version of $SkewScore_y(p)$ is smaller or larger than some threshold?

---

> ### Author Response · Authors · 2024-11-21
>
> After obtaining the sample version of $\textit{SkewScore}_ {x_ j}(p)$, we identify the variable $x_ j$ with the minimum estimated $\textit{SkewScore}_ {x_ j}(p)$ to be the leaf node (sink node), as detailed in Line 6 of Algorithm 1 in the manuscript. This leaf node represents the variable that is most likely to be causally downstream, having the least skewness in the score's projection. We then remove this leaf node from the set of nodes (Lines 8-9 of Algorithm 1) and repeat the process for the remaining variables. By iteratively removing the node with the smallest $\textit{SkewScore}$, we establish a topological order that reflects the causal direction among the variables.
>
> To further clarify, we do not determine the causal direction based on whether the sample version of $\textit{SkewScore}_ {x_ j}(p)$ is larger or smaller than a specific threshold. Instead, we employ the above process, which provides a systematic determination of the causal order. Thank you for your question.

---

> ### Author Response · Authors · 2024-11-29
> **Looking forward to discussion**
>
> Dear Reviewer 8P9r,
>
> Thank you for your efforts in reviewing this paper. We have provided responses to your further comments. Could you please check whether the responses properly addressed your concerns, or if you have further comments? Your feedback would be appreciated. Thanks a lot for your time.
>
> Best regards,
>
> Authors of Paper 9319

---

### Official Review · Reviewer_b2UL · 2024-11-08

**Soundness:** 2
**Presentation:** 3
**Contribution:** 2
**Rating:** 6
**Confidence:** 4

**Summary:**

This work proposes a causal discovery method for a class of restricted structural causal models with heteroscedastic symmetric noise. The identification of the causal graph is established using a score that quantifies the skewness of the score function. A causal discovery algorithm is then proposed based on the skewness score.

**Strengths:**

The notation is well-organized and the paper is easy to follow overall.

**Weaknesses:**

1. Fact 2 is incorrect. Let X follow univariate exponential distribution with parameter $\lambda$, then $E[\frac{d \text{log} p(X)} { dx}] = -\lambda$. Its proof is also incorrect.

2. The proof of Theorem 1 heavily relies on the assumption of symmetric noise. Without this symmetry, skewness no longer provides useful implications for the causal direction, as the proof of Theorem 1 breaks down.

3. Definition 1 needs detailed discussions.  As stated, "this measure captures the asymmetry...," it would be helpful to demonstrate that it is zero when the distribution is symmetric.

**Questions:**

Is it possible to provide identification results such as 'the skew score is larger under the wrong causal direction' without symmetric noise assumption?

---

> ### Author Response · Authors · 2024-11-18
>
> > Fact 2 is incorrect. Let $X$ follow a univariate exponential distribution with parameter $\lambda$, then $\mathbb{E}\left[\frac{d \log p(X)}{dx}\right] = -\lambda$. Its proof is also incorrect.
>
> Fact 2 is **correct** under the assumption $p_ X(x)\in C^1(\mathbb{R}^d,\mathbb{R}_ +)$ stated in Line 150 of the original manuscript, where $ C^1(\mathbb{R}^d, \mathbb{R}_ +) $ is the space of continuously differentiable functions from $\mathbb{R}^d$ to the positive real numbers $\mathbb{R}_ +$ (Line 117). The exponential distribution does not satisfy this assumption due to the singularity at $x=0$, where $p(0-)=0$ and $p(0+)=1$.
>
> The proof of Fact 2 is based on the Newton-Leibniz formula. In the last step of the proof, since $p$ is continuously differentiable, by the Newton-Leibniz formula: $$\int_ {\mathbb{R}}\frac{\partial}{\partial x_ i}p(x_ 1,x_ 2\cdots,x_ d) d x_ i = p(x_ 1,x_ 2\cdots,x_ d) |^{x_ i=+\infty}_ {x_ i = -\infty} = 0-0 = 0,$$
> where $\lim_ {x_ i\rightarrow+\infty}p(x_ 1,x_ 2\cdots,x_ d) =\lim_ {x_ i\rightarrow -\infty}p(x_ 1,x_ 2\cdots,x_ d) =0 $ because $p$ is a probability density.
>
> We also note that the assumption $p_ X(x)\in C^1(\mathbb{R}^d,\mathbb{R}_ +)$ can be relaxed. For example, Fact 2 holds for uniform distribution on $[-1,1]$ despite having singularities at $-1$ and $1$, as the densities at these two endpoints cancel out after the Newton-Leibniz step.
>
> We have included these derivations and discussion in Appendix D of the revised manuscript. We have also added your example to help readers understand Fact 2 more comprehensively. Thanks for the question.
>
>
> > The proof of Theorem 1 heavily relies on the assumption of symmetric noise. Without this symmetry, skewness no longer provides useful implications for the causal direction, as the proof of Theorem 1 breaks down.
>
> Thank you for highlighting the reliance of Theorem 1 on the assumption of symmetric noise. We would like to emphasize the following two points.
>
> (1) We believe that the symmetric noise assumption is a reasonable generalization of the Gaussian noise assumption utilized in previous work on multivariate heteroscedastic causal models (Duong \& Nguyen, 2023). Symmetric noise distributions cover a wide range of commonly used noise types, including Gaussian and Student’s t-distributions, as mentioned in Lines 139-142 of our manuscript.
>
> (2) To evaluate the robustness of SkewScore under slightly skewed noise distributions, we conducted experiments using Gumbel noise in the bivariate heteroscedastic noise model (see Appendix J). The results indicate that while there is a decrease in accuracy, SkewScore’s performance degradation is less significant compared to HOST (Duong \& Nguyen, 2023). This suggests that SkewScore may retain effectiveness even when the noise deviates from perfect symmetry.
>
> We agree that extending our theoretical framework to accommodate non-symmetric noise distributions is both important and achievable. We will discuss this further in our response to your next question.
>
>
> > Is it possible to provide identification results such as "the skew score is larger under the wrong causal direction" without the symmetric noise assumption?
>
> Thank you for raising this insightful question. Providing identification results without the symmetric noise assumption is indeed more challenging, but it is possible under certain conditions. Specifically, if the asymmetry introduced by the link functions  $f$  (conditional mean) and  $\sigma$  (conditional standard deviation) is sufficiently strong compared to the asymmetry of the exogenous noise, similar identification results can be achieved.
> One potential approach is to quantify the asymmetry of the exogenous noise using measures like skewness or the SkewScore. By doing so, we can establish sufficient conditions on the link functions that account for the noise’s skewness. This would likely involve developing an inequality version of Assumption 1 that incorporates both the properties of the link functions and the skewness of the exogenous noise. We are optimistic that with further research, we can relax the symmetric noise assumption and provide more general identification results.
>
>
> > Definition 1 needs detailed discussions. As stated, "this measure captures the asymmetry...," it would be helpful to demonstrate that it is zero when the distribution is symmetric.
>
> Thank you for your suggestion. We have included additional discussions in Appendix A in the revision. We demonstrate that the SkewScore measure is zero when the conditional density is symmetric. Additionally, we provide two examples calculating the SkewScore for skewed distributions. Along with the original manuscript's discussion on the motivation for considering the SkewScore measure, we believe these enhancements will improve the reader's understanding of this definition.
>
> Reference:
>
> Duong, B., & Nguyen, T. (2023). Heteroscedastic Causal Structure Learning. In ECAI 2023 (pp. 598-605). IOS Press.

---

> > ### Comment · Reviewer_b2UL · 2024-11-21
> > **reply**
> >
> > 1. Regarding Fact 2, a function $f \in C^{1}(\mathbb{R}^{d}, \mathbb{R}_{+})$ can have a support that is a subset of $\mathbb{R}^{d}$ and its function values outside the support are undefined (e.g., the exponential distribution example). My point is that the assumption should be revised to "the density function is strictly positive on $\mathbb{R}^{d}$".
> >
> > 2. If the identification results cannot be extended to general classes of non-symmetrical noise, then there is a gap between your theory and algorithm. The algorithm uses the skewness score as a metric to identify sink nodes but never checks if the score is close to zero. If the score is consistently "not close" to zero, then the algorithm may fail completely since there are no identification results for such a setting.

---

> ### Author Response · Authors · 2024-11-22
>
> 1. Thank you for this suggestion. At the same time, we would like to point out that $f\in C^1(\mathbb{R}^d, \mathbb{R}_ +)$ inherently implies that the $f$ is strictly positive on $\mathbb{R}^d$, because it means that $f$ maps every point in the domain $\mathbb{R}^d$ to the codomain $\mathbb{R}_ +=(0,\infty)$. The definitions of $\mathbb{R}_ +$ and $C^1(\mathbb{R}^d, \mathbb{R}_ +)$ are outlined in Lines 116-118 of the manuscript. For the exponential distribution example, we can state that $f_ {exp} \in C^1([0,\infty), \mathbb{R}_ +)$ or $f_ {exp} \in \textrm{piecewise} \ C^1(\mathbb{R}, \mathbb{R}_ + \cup \\{0\\})$, but $f_ {exp} \notin C^1(\mathbb{R}, \mathbb{R}_+)$. However, in light of your concern and to avoid any confusion, we have emphasized that "the density function is strictly positive on $\mathbb{R}^d$" in Line 151 of the revision. Thanks again for this comment, which helps improve the clarity.
>
> 2. Thank you for your suggestion. It is a good idea to check whether the model satisfies our assumption by checking if the smallest SkewScore is close to zero. Potential approaches for this verification are non-parametric hypothesis testing methods such as bootstrapping and permutation tests, since we do not assume a specific type of noise (e.g., Gaussian). We have included the discussion on checking the model assumption in Lines 277-280 of the revision. While we acknowledge the value of incorporating such a verification step, we believe this challenge is inherent to most causal discovery results and does not represent a major weakness or one specific to our paper.

---

> > ### Comment · Reviewer_b2UL · 2024-11-25
> > **reply**
> >
> > This notation is a minor issue. You are likely correct regarding the support.
> >
> > The main issue remains the disconnect between the identification results and the algorithm. The identification results rely heavily on the structure of model (3) with symmetric additive noise, yet your algorithm neither fits model (3) nor checks for symmetry. In fact, I believe the algorithm’s performance might suffer if it were constrained to fit the model or enforce symmetry. Since the current algorithm performs well on real data, my concern is that the identification results are too restrictive. Nonetheless, I will slightly increase my score.

---

> > > ### Author Response · Authors · 2024-11-26
> > > **Thank you for your increased score and support**
> > >
> > > Thank you for your thoughtful feedback. Deriving weaker assumptions to identify functional causal models is an important direction for our future work (and possibly for the field in general). We greatly appreciate your increased score and support.

---

### Official Review · Reviewer_h4sn · 2024-11-08

**Soundness:** 4
**Presentation:** 3
**Contribution:** 3
**Rating:** 6
**Confidence:** 4

**Summary:**

This paper provides a novel causal discovery algorithm that leverages skewness to determine causal direction when heteroscedastic noise (HN) is present.

**Strengths:**

1. An example in Section 5 provides a clear, practical illustration of the algorithm, connecting theoretical concepts with application.

2. This paper provides a well-structured and self-contained summary of prior works in causal discovery algorithms based on functional causal models, allowing readers to clearly follow the evolution for learning causal graphs. As a reviewer, this summary enables me to track the advancements in the field and understand the recent progress.

3. The core idea behind the proposed method is simple and intuitive, based on the asymmetry of score skewness to identify causal directions. This approach is conceptually accessible and computationally efficient solutions.

4. The experiments provide strong empirical evidences of the paper’s contributions, with a wide range of scenarios that demonstrate the robustness of the proposed framework in heteroscedastic settings.

**Weaknesses:**

1. The paper provides correctness guarantees only for the two-variable case, where skewness in the score function can reliably distinguish causal directions. However, for settings with more than two variables, there is no formal guarantee of correctness for Algorithm 1.

2. Current empirical results seem insufficient in showcasing its performance in high-dimensional scenarios. In other words, it remains unclear whether SkewScore scales effectively as the number of variables grows significantly.

3. While the authors claim robustness in the presence of latent confounders, this robustness appears to be limited to simple bivariate cases. In multivariate settings, SkewScore can still produce incorrect causal graphs when latent confounders are present. For instance, in the causal structure $X_1 \rightarrow X_2 \rightarrow X_3$ with a latent confounder $L$ influencing both $X_1$ and $X_3$ ($X_1 \leftarrow L \rightarrow X_3$), the method would infer the structure $X_1 \rightarrow X_2 \rightarrow X_3$ along with a spurious direct link $X_1 \rightarrow X_3$. This incorrect edge arises because the latent confounder $L$ induces a conditional dependence between $X_1$ and $X_3$, which SkewScore interprets as a direct causal link. As a result, while the method can handle some latent confounding, it does not offer sufficient mechanisms to detect and mitigate confounding in more complex structures, limiting its application in practical scenarios where latent variables are common.

**Questions:**

1. Is the algorithm scalable?  It would be valuable to discuss if this approach can handle high-dimensional datasets effectively and whether its complexity scales polynomially or exponentially with the number of variables.

2. Can you discuss the relation between the proposed method with other types of causal discovery algorithms, such as (1) constraints-based algorithms such as PC algorithm etc., and (2) continuous optimization based algorithms such as the one in https://arxiv.org/abs/1803.01422, which provides scalability?

3. What are the effects of small sample sizes on SkewScore’s performance? It would be great if authors can discuss the convergence guarantee of the algorithm in terms of the size of samples.

4. Can the proposed algorithm provide confidence intervals or uncertainty quantification in measuring the skewness or determining causal directionality?

---

> ### Author Response · Authors · 2024-11-21
>
> > Correctness guarantees for the multi-variable case
>
> Thank you for you question. The guarantee for multivariate settings is provided in Corollary 2 (Lines 256-265 of the manuscript), which is an extension of Theorem 1.
>
> > Latent Confounders in multivariate settings: incorrect causal graphs. For $X_ 1 \to X_ 2 \to X_ 3$ with a latent confounder $L$ influencing both $X_ 1$ and $X_ 3$ ($X_ 1 \leftarrow L \to X_ 3$), the method would infer the structure $X_ 1 \to X_ 3$ along with a spurious direct link $X_ 1 \to X_ 3$.
>
> Thank you for your insightful question on the latent confounders in multivariate settings. Our current procedure first infers the topological ordering with the SkewScore criterion, and then uses conditional independence tests to further remove edges and identify the DAG. We agree that the way we perform conditional independence tests will lead to spurious direct links when there are latent confounders in the multivariate setting. A potential way to mitigate this issue is to leverage the idea from FCI, which will involve more sophisticated ways of performing conditional independence tests (after obtaining the topological ordering using our SkewScore criterion) and output the Partial Ancestral Graph (PAG) instead of just a DAG. Note that this is an interesting direction and is beyond the scope of our work, which is left for future investigation. We have included this discussion in the revision (Lines 356-360) and hope it addresses your concern.
>
>
>
> > Scalability; complexity scales polynomially or exponentially with the number of variables.
>
> Thank you for your question. The computational complexity is $O(d^2)$ with respect to the number of variables $d$. SkewScore can be divided into two phases: the causal order recovery with the SkewScore criterion and the DAG recovery with independence tests. The first phase includes $d$ iterations that perform SSM, where SSM scales to high dimensional data with Adam optimizer (Song et al., 2020). The second phase includes $O(d^2)$ conditional independence tests.
>
> We acknowledge that further exploration is required to fully extend SkewScore to high-dimensional cases. Future work will focus on improving its scalability to enhance its applicability.
>
> >  Relation between (1) constraints-based algorithms such as the PC algorithm, and (2) continuous optimization-based algorithms such as the one in \url{https://arxiv.org/abs/1803.01422}, which provides scalability?
>
> (1) The PC algorithm runs in exponential time in $d$ (the number of variables) in the worst case, since it performs the CI tests for all subsets of variables in the skeleton recovery step. SkewScore reduces this computation complexity by first estimating the topological order, which constrains the DAG to be a sub-graph of a certain fully connected DAG, and then performs only $O(d^2)$ CI tests. The PC algorithm imposes fewer assumptions than SkewScore, as it does not rely on a functional causal model, and thus it can only identify the underlying structure up to Markov equivalence class but not the complete DAG.
>
> (2) Continuous optimization based methods typically also make certain assumptions on the functional causal model, such as NOTEARS that assumes a linear structural equation model with equal noise variances, NOTEARS-MLP that assumes additive noise models with equal noise variances, and GraN-DAG that assumes additive noise models with Gaussian noises. Although these methods are relatively computationally efficient, recent studies have shown that they may exploit certain type of artifacts of simulated data (Reisach et al., 2021) and may be susceptible to the nonconvexity issue (Ng et al., 2024), thus limiting their applicability.
>
> Summary: The PC algorithm imposes fewer assumptions but incurs higher computational costs and only identifies the Markov equivalence class. Continuous optimization based methods are computationally efficient but have several practice issues. SkewScore provides a balanced tradeoff, combining moderate computational complexity with broader applicability.

---

> ### Author Response · Authors · 2024-11-21
>
> >  Small sample sizes
>
> Thanks for your question. We conducted the experiment on the effect of sample size (see Table 5 in Appendix J of the manuscript). Our method becomes more accurate with a larger sample size. Even with smaller sample sizes, SkewScore shows strong performance, maintaining a reliable level of accuracy.
>
> On the theoretical side, the asymptotic normality of the SSM estimator is established under appropriate assumptions in Theorem 3 of [Song et al., 2020]. However, we acknowledge that deriving nonasymptotic guarantees under general assumptions that cover heteroscedastic causal models remains an open challenge. A limitation is the Lipschitzness of the score function assumed in previous work [Zhu et al., 2024], which restricts the family of causal models SkewScore can address. Future work will aim to close this gap by extending existing results, such as those in [Zhu et al., 2024], to accommodate more general, non-Gaussian scenarios.
>
> > Confidence intervals or uncertainty quantification in measuring the skewness or determining causal directionality
>
> Yes, we can perform hypothesis testing for skewness $\neq 0$. Since we do not assume a specific type of noise (e.g., Gaussian), non-parametric methods such as bootstrapping and permutation tests could be employed. Thank you for your question.
>
>
> References:
>
> Song, Y., Garg, S., Shi, J., \& Ermon, S. (2020, August). Sliced score matching: A scalable approach to density and score estimation. In Uncertainty in Artificial Intelligence (pp. 574-584). PMLR.
>
> Spirtes, P., \& Glymour, C. (1991). An algorithm for fast recovery of sparse causal graphs. Social science computer review, 9(1), 62-72.
>
> Zheng, X., Aragam, B., Ravikumar, P. K., \& Xing, E. P. (2018). Dags with no tears: Continuous optimization for structure learning. Advances in neural information processing systems, 31.
>
> Spirtes, P., Glymour, C., \& Scheines, R. (2001). Causation, prediction, and search. MIT press.
>
> Reisach, A., Seiler, C., \& Weichwald, S. (2021). Beware of the simulated dag! causal discovery benchmarks may be easy to game. Advances in Neural Information Processing Systems, 34, 27772-27784.
>
> Ng, I., Huang, B., \& Zhang, K. (2024, March). Structure learning with continuous optimization: A sober look and beyond. In Causal Learning and Reasoning (pp. 71-105). PMLR.

---

> ### Author Response · Authors · 2024-12-01
> **Looking forward to discussion**
>
> Dear Reviewer h4sn,
>
> Thank you for your time and effort in reviewing our paper. We have carefully addressed your concerns in the rebuttal and would greatly appreciate it if you could review our response. If you have further questions or comments after reading the rebuttal, we hope to have the opportunity to address them.
>
> Thanks again for your time and consideration.
>
> Best regards,
> Authors of Paper 9319

---

### Meta-Review · Area_Chair_czDQ · 2024-12-20

**Metareview:**

The paper tackles an important and challenging problem of causal discovery with heteroscedastic noise (HN), by utilising the third moment (skewness) of the score function (derivative of the log density). The idea with contrastive-based learning idea for the HN case is novel and substantial; the connection with causal discovery, score matching and skewness further strengthen the insights of the proposed approach. As the reviewers generally reach the consensus that the paper is nicely-organised and well-written; the contribution is significant for our community, the paper is recommended to appear and highlight in the ICLR conference.

**Additional Comments On Reviewer Discussion:**

The discussion has been fruitful and resolve most of reviewers' concerns.

---

### Decision · Program_Chairs · 2025-01-22

Accept (Poster)